# Understanding and Improving Transfer Learning of Deep Models via Neural Collapse

**Xiao Li**[*]                                                                      *xlxiao@umich.edu*
*Department of Electrical Engineering and Computer Science, University of Michigan*

**Sheng Liu**[*]                                                                     *shengl@stanford.edu*
*Biomedical Data Sciences, Stanford University*

**Jinxin Zhou**                                                         *zhou.3820@buckeyemail.osu.edu*
*Department of Computer Science and Engineering, Ohio State University*

**Xinyu Lu**                                                                    *xinyulu0107@gmail.com*
*School of Computer Science, Carnegie Mellon University*

**Carlos Fernandez-Granda**                                                 *cfgranda@cims.nyu.edu*
*Center for Data Science, New York University*

**Zhihui Zhu**                                                                      *zhu.3440@osu.edu*
*Department of Computer Science and Engineering, Ohio State University*

**Qing Qu**                                                                        *qingqu@umich.edu*
*Department of Electrical Engineering and Computer Science, University of Michigan*

**Reviewed on OpenReview:** `https://openreview.net/forum?id=o8r84MzTQB`

## Abstract

With the ever-increasing complexity of large-scale pre-trained models coupled with a shortage of labeled data for downstream training, transfer learning has become the primary approach in many fields, including natural language processing, computer vision, and multimodal learning. Despite recent progress, the fine-tuning process for large-scale pre-trained models in vision still mostly relies on trial and error. This work investigates the relationship between neural collapse (NC) and transfer learning for classification problems. NC is an intriguing while prevalent phenomenon that has been recently discovered in terms of the final-layer features and linear classifiers of trained neural networks. Specifically, during the terminal phase of training, NC implies that the variability of the features within each class diminishes to zero, while the means of features between classes are maximally and equally distanced. In this work, we examine the NC attributes of pre-trained models on both downstream and training data for transfer learning, and we find strong correlation between feature collapse and downstream performance. In particular, we discovered a systematic pattern that emerges when linear probing pre-trained models on downstream training data: the more feature collapse of pre-trained models on downstream data, the higher the transfer accuracy. Additionally, we also studied the relationship between NC and transfer accuracy on the training data. Moreover, these findings allow us to develop a principled, parameter-efficient fine-tuning method that employs skip-connection to induce the last-layer feature collapse on downstream data. Our proposed fine-tuning methods deliver

---

[*]The first two authors contributed equally to the work.

good performances while reducing fine-tuning parameters by at least 90% and mitigating overfitting in situations especially when the downstream data is scarce.

## 1 Introduction

Transfer learning has gained widespread popularity in computer vision, medical imaging, and natural language processing (Zhuang et al., 2020; Devlin et al., 2019; Cheplygina et al., 2019). By taking advantage of domain similarity, a pre-trained, large model from source datasets is reused as a starting point or feature extractor for fine-tuning a new model on a smaller downstream task (Zhuang et al., 2020). The reuse of the pre-trained model during fine-tuning reduces computational costs significantly and leads to superior performance on problems with limited training data. Despite of recent advances, the underlying mechanism of transfer learning is still far from understood, and many of existing approaches lack principled guidance.

In the past, the bulk of research has focused on improving data representations during the phase of model pre-training, with various metrics developed to assess representation quality on source data, such as validation accuracy on ImageNet (Kornblith et al., 2019) and feature diversity (Kornblith et al., 2021; Nayman et al., 2022). However, the evaluation of representation quality has become increasingly challenging due to the scale and limited accessibility of source data. Additionally, with the emergence of large-scale foundation models (Bommasani et al., 2021; Zhou et al., 2023) such as GPT-4 (OpenAI, 2023), SAM (Kirillov et al., 2023), and CLIP (Radford et al., 2021), the research focus has shifted towards the evaluating and fine-tuning of these pre-trained foundation models (Kumar et al., 2022) on downstream tasks.

Therefore, in this work, we mainly focus on the downstream data for evaluating and fine-tuning large-scale pre-trained models. We examine the principles of transfer learning by drawing connections to an intriguing phenomenon prevalent across different network architectures and datasets, termed '*Neural Collapse*" ($\mathcal{NC}$) (Papyan et al., 2020; Han et al., 2022), where the last-layer features and classifiers "collapse" to simple while elegant mathematical structures on the training data. Specifically, during the terminal phase of training, it has been observed that (*i*) the within-class variability of last-layer features collapses to zero for each class and (*ii*) the between-class means and last-layer classifiers collapse to the vertices of a Simplex Equiangular Tight Frame (ETF) up to scaling.

While the initial discovery of $\mathcal{NC}$ was rooted in the analysis of the source training data, our study demonstrates a compelling relationship between $\mathcal{NC}$ and transferability on downstream datasets. We adapt the metrics for assessing $\mathcal{NC}$ to gauge the quality of learned representations concerning both *within-class diversity* and *between-class discrimination*. Remarkably, when applying linear probing to pre-trained models on downstream training data, we uncover an intriguing trend: as the features of pre-trained models collapse more significantly on the downstream training data, the transfer accuracy improves. Capitalizing on this insight, we design more principled and parameter-efficient fine-tuning approaches for large-scale pre-trained models, including foundation models like CLIP (Dosovitskiy et al., 2021; Radford et al., 2021).

Finally, for a more comprehensive study, we also investigate the relationship between transferability and $\mathcal{NC}$ metrics on the source training data. We considered different types of factors that affect transfer accuracy of pre-trained models, such as training losses, projection heads, and network architectures. Our work reveals the limitation of using source data for predicting transfer accuracy.

**Contributions.** In summary, we highlight our contributions below.

- **More collapsed features on downstream tasks lead to better transfer accuracy.** Through an extensive examination of pre-trained models using linear probing across various scenarios, our work shows the following relationship: a higher degree of feature collapse on downstream data tends to yield improved transfer accuracy. This phenomenon is supported by comprehensive experiments conducted on multiple downstream datasets (Krizhevsky et al., 2009; Maji et al., 2013; Cimpoi et al., 2014; Parkhi et al., 2012), diverse pre-trained models (He et al., 2016; Dosovitskiy et al., 2021; Huang et al., 2017; Sandler et al., 2018; Radford et al., 2021), and even within the context of few-shot learning (Tian et al., 2020; Liu et al., 2023b). Notably, we find that this relationship holds across different layers when linear probing is applied to features of distinct layers from the same pre-trained model, thereby offering valuable insights for the principled design of more effective linear probing techniques.

- **More efficient fine-tuning of large pre-trained models via $\mathcal{NC}$.** Based on the aforementioned insights, we propose a simple and memory-efficient fine-tuning approach that achieves good results in vision classification tasks. Our method revolves around the utilization of skip connections to fine-tune a critical layer in the pre-trained network, thereby maximizing the collapse of the last-layer feature. To validate the effectiveness of our strategy, we conduct experiments using publicly available pre-trained models, including ResNet, ViT, and CLIP (He et al., 2016; Dosovitskiy et al., 2021; Radford et al., 2021). Remarkably, our method substantially reduces the number of fine-tuning parameters by at least 90% compared to full model fine-tuning. Moreover, in the context of low-shot learning, our approach exhibits improved robustness to data scarcity, exhibiting reduced overfitting tendencies.
- **Limitations of studying transfer accuracy using source training data.** In addition to investigating the connection between downstream $\mathcal{NC}$ and transferability, we also studied the relationship between source $\mathcal{NC}$ and transferability. More specifically, we investigated different key factors that affect the transfer accuracy based upon the $\mathcal{NC}$ metrics on the source data, such as loss functions (Hui & Belkin, 2020), the projection head (Chen et al., 2020), data augmentations (Chen & He, 2021; Khosla et al., 2020), and adversarial training (Salman et al., 2020; Deng et al., 2021). Within a certain threshold, we found that the more diverse the features are, the better the transferability of the pre-trained model. However, as randomly generated features that are diverse do not generalize well, the relationship does not always hold and using source data diversity metrics has limitations for predicting transfer accuracy.

The rest of the paper is structured as follows. In Section 2, we briefly review the concept of Neural Collapse ($\mathcal{NC}$) and introduce our method for measuring transferability through $\mathcal{NC}$. In Section 3, we demonstrate how $\mathcal{NC}$ can be used to evaluate a pre-trained model's transferability on a downstream dataset prior to fine-tuning. Motivated by this observation, we propose parameter-efficient fine-tuning methods in Section 4, which show robust performance in both standard and limited data scenarios. In Section 5, we extend our study to the relationship between $\mathcal{NC}$ and transferability on the source dataset, revealing a more complex and non-monotonic relationship. Finally, we conclude the paper in Section 6. Discussions of related work, detailed experimental setups and additional experiments are postponed to the Appendix.

## 2 Evaluating Pre-trained Models via NC

In this section, we provide a brief overview of the $\mathcal{NC}$ phenomenon, followed by the introduction of metrics for evaluating the quality of learned representations for transfer learning.

**Basics of Deep Neural Networks.** Let us first introduce some basic notations by considering a multi-class (e.g., $K$ class) classification problem with finite training samples. Let $\{n_k\}_{k=1}^{K}$ be the number of training samples in each class. Let $\boldsymbol{x}_{k,i}$ denote the $i$th input data in the $k$th class ($i \in [n_k]$, $k \in [K]$), and the corresponding one-hot training label is represented by $\mathbf{y}_k \in \mathbb{R}^K$, with only the $k$th entry equal to 1. Thus, given any input data $\boldsymbol{x}_{k,i}$, we learn a deep network to fit the corresponding (one-hot) training label $\boldsymbol{y}_k$ such that

$$\boldsymbol{y}_k \approx \psi_{\boldsymbol{\Theta}}(\boldsymbol{x}_{k,i}) = \underbrace{\boldsymbol{W}_L}_{\textbf{linear classifier } \boldsymbol{W}} \cdot \sigma \left( \boldsymbol{W}_{L-1} \cdots \sigma \underbrace{\left( \boldsymbol{W}_1 \boldsymbol{x}_{k,i} + \boldsymbol{b}_1 \right) + \boldsymbol{b}_{L-1} \right)}_{\textbf{feature } \boldsymbol{h}_{k,i} = \phi_{\boldsymbol{\theta}}(\boldsymbol{x}_{k,i})} + \boldsymbol{b}_L, \tag{1}$$

where $\boldsymbol{W} = \boldsymbol{W}_L$ represents the last-layer linear classifier and $\boldsymbol{h}_{k,i} = \boldsymbol{h}(\boldsymbol{x}_{k,i}) = \phi_{\boldsymbol{\theta}}(\boldsymbol{x}_{k,i})$ is a deep hierarchical representation (or feature) of the input $\boldsymbol{x}_{k,i}$. Here, for a $L$-layer deep network $\psi_{\boldsymbol{\Theta}}(\boldsymbol{x})$, each layer is composed of an affine transformation, followed by a nonlinear activation $\sigma(\cdot)$ and normalization functions (e.g., BatchNorm (Ioffe & Szegedy, 2015)). We use $\boldsymbol{\Theta}$ to denote all the network parameters of $\psi_{\boldsymbol{\Theta}}(\boldsymbol{x})$ and $\boldsymbol{\theta}$ to denote the network parameters of $\phi_{\boldsymbol{\theta}}(\boldsymbol{x})$. Additionally, we use

$$\boldsymbol{H} = \begin{bmatrix} \boldsymbol{H}_1 & \boldsymbol{H}_2 & \cdots & \boldsymbol{H}_K \end{bmatrix} \in \mathbb{R}^{d \times N}, \quad \boldsymbol{H}_k = \begin{bmatrix} \boldsymbol{h}_{k,1} & \cdots & \boldsymbol{h}_{k,n} \end{bmatrix} \in \mathbb{R}^{d \times n}, \ \forall \, k \in [K],$$

to represent all the features in matrix form. Additionally, the class mean for each class is written as

$$\overline{\boldsymbol{H}} := \begin{bmatrix} \overline{\boldsymbol{h}}_1 & \cdots & \overline{\boldsymbol{h}}_K \end{bmatrix} \in \mathbb{R}^{d \times K} \quad \text{and} \quad \overline{\boldsymbol{h}}_k := \frac{1}{n_k} \sum_{i=1}^{n_k} \boldsymbol{h}_{k,i}, \quad ; \forall \, k \in [K].$$

Accordingly, we denote the global mean of $\boldsymbol{H}$ as $\boldsymbol{h}_G = \frac{1}{K} \sum_{k=1}^{N} \overline{\boldsymbol{h}}_k$.

**A Review of Neural Collapse.** It has been widely observed that the last-layer features $\mathbf{H}$ and classifiers $\mathbf{W}$ of a trained network on a balanced training dataset $\{\boldsymbol{x}_{k,i}, \boldsymbol{y}_k\}$ with $n = n_1 = n_2 = \cdots = n_K$ exhibit simple but elegant mathematical structures (Papyan et al., 2020; Papyan, 2020). Here, we highlight two key properties below.[1]

- **Within-class variability collapse.** For each class, the last-layer features collapse to their means,

$$\boldsymbol{h}_{k,i} \to \overline{\boldsymbol{h}}_k, \quad \forall\, 1 \le i \le n,\ 1 \le k \le K. \tag{2}$$

- **Maximum between-class separation.** The class-means $\overline{\boldsymbol{h}}_k$ centered at their global mean $\mathbf{h}_G$ are not only linearly separable but also maximally distant and form a Simplex Equiangular Tight Frame (ETF): for some $c > 0$, $\overline{\boldsymbol{H}} = \begin{bmatrix} \overline{\boldsymbol{h}}_1 - \boldsymbol{h}_G & \cdots & \overline{\boldsymbol{h}}_K - \boldsymbol{h}_G \end{bmatrix}$ satisfies

$$\overline{\boldsymbol{H}}^\top \overline{\boldsymbol{H}} \;=\; \frac{cK}{K-1}\left( \boldsymbol{I}_K - \frac{1}{K}\mathbf{1}_K\mathbf{1}_K^\top \right). \tag{3}$$

Recent studies have shown that $\mathcal{NC}$ is prevalent across a wide range of classification problems (Papyan et al., 2020; Mixon et al., 2020; Zhou et al., 2022; Han et al., 2022; Fang et al., 2021; Graf et al., 2021; Zhou et al.) on the source training data, regardless of the loss function used, the neural network architecture and the dataset. Intuitively, the prevalence of $\mathcal{NC}$ phenomenon implies that the features in each class are maximally separated on the training data, and the network learns a max-margin linear classifier in the last-layer. Additionally, if $\mathcal{NC}$ would also occur on downstream data, it could serve as an indicator of the transferability of pre-trained models that we study below.

**Measuring the Transferability of Pre-trained Models via $\mathcal{NC}$ Metrics.** Based on the above discussion, we can assess the transferability of pre-trained models on the down by measuring the feature diversity and separation using metrics for evaluating $\mathcal{NC}$ (Papyan et al., 2020; Zhu et al., 2021) defined as follows:

$$\mathcal{NC}_1 \;:=\; \frac{1}{K}\, \mathrm{trace}\left( \boldsymbol{\Sigma}_W \boldsymbol{\Sigma}_B^\dagger \right). \tag{4}$$

More specifically, it measures the magnitude of the within-class covariance matrix $\boldsymbol{\Sigma}_W \in \mathbb{R}^{d \times d}$ of the learned features compared to the between-class covariance matrix $\boldsymbol{\Sigma}_B \in \mathbb{R}^{d \times d}$, where

$$\boldsymbol{\Sigma}_W \;:=\; \frac{1}{nK}\sum_{k=1}^{K}\sum_{i=1}^{n}\left(\boldsymbol{h}_{k,i} - \overline{\boldsymbol{h}}_k\right)\left(\boldsymbol{h}_{k,i} - \overline{\boldsymbol{h}}_k\right)^\top, \quad \boldsymbol{\Sigma}_B \;:=\; \frac{1}{K}\sum_{k=1}^{K}\left(\overline{\boldsymbol{h}}_k - \boldsymbol{h}_G\right)\left(\overline{\boldsymbol{h}}_k - \boldsymbol{h}_G\right)^\top.$$

Here, $\boldsymbol{\Sigma}_B^\dagger$ represents the pseudo-inverse of $\boldsymbol{\Sigma}_B$, which normalizes $\boldsymbol{\Sigma}_W$ to capture the relative relationship between the two covariances. Essentially, if the features of each class are more closely clustered around their corresponding class means, $\mathcal{NC}_1$ will be smaller. Conversely, if the class means are more separated, $\mathcal{NC}_1$ will also be smaller for the same $\boldsymbol{\Sigma}_W$. However, computing this metric in (4) can be computationally expensive due to the pseudo-inverse, which is often the case for large models. To address this issue, alternative metrics such as class-distance normalized variance (CDNV) (Galanti et al., 2022b) and numerical rank (Zhou et al., 2022) have been introduced recently; we refer interested readers to Appendix D for more details.

Based upon the above, in the following we propose to evaluate the quality of learned representations for transfer learning using the metric in equation 4 on the *downstream/source data*, providing new insights into model fine-tuning. Specifically, in Section 3, we focus on downstream tasks and find that the $\mathcal{NC}$ metric measured on the downstream dataset and transfer accuracy are negatively correlated: smaller $\mathcal{NC}_1$ on downstream data leads to better transfer accuracy. Based on the findings, we will propose a simple and parameter efficient fine-tuning method in Section 4 that works well in transfer learning. Additionally, we also investigate the relationship between $\mathcal{NC}$ on the source dataset and pre-training in Section 5. The experimental setup is detailed in Appendix E. In order to comprehensively understand the relationship between $\mathcal{NC}$ and transfer learning, we measure $\mathcal{NC}$ at different layers of various models as different sections tend to have different focuses, see Table 1 for the setup of each section.

---

[1]Additionally, self-duality convergence has also been observed in the sense that $\boldsymbol{w}_k = c'\overline{\boldsymbol{h}}_k$ for some $c' > 0$. We omit them here because they are not the main focus of this work.

Table 1: **Genenral experimental setup for each section.** We study the relationship between $\mathcal{NC}$ and transfer learning with different setups for each section.

| | Fine-tuned/Pretrained Models | Dataset for Evaluation of $\mathcal{NC}$ | Using Intermediate Layer $\mathcal{NC}$ | Using Penultimate Layer $\mathcal{NC}$ |
|---|---|---|---|---|
| Section 3 | Pre-trained Models | Downstream training data | Fig. 3 | Fig. 1,2,4; Table 2 |
| Section 4 | Fine-tuned Models | Downstream training data | - | Fig. 7; Table 3 |
| Section 5 | Pre-trained Models | Source training data | - | Fig. 10 |

Figure 1: **Transfer accuracy and $\mathcal{NC}_1$ of Cifar-100 pre-trained models on different downstream tasks.** We pre-train ResNet50 models on Cifar-100 using different levels of data augmentation or adversarial training. Here, $\mathcal{NC}_1$ is measured on the downstream Cifar-10 dataset.

## 3 Study of $\mathcal{NC}$ & Transfer Accuracy without Model Fine-tuning

First, we explore the relationship between the $\mathcal{NC}_1$ metric and transfer accuracy by evaluating pre-trained models on downstream data **before fine-tuning**. The practice of transferring pre-trained large models to smaller downstream tasks has become a common approach in NLP (Devlin et al., 2019; Houlsby et al., 2019; Hu et al., 2021), vision (Dosovitskiy et al., 2021; Evci et al., 2022; Adler et al., 2020), and multi-modal learning. In the meanwhile, evaluating $\mathcal{NC}$ metrics of pre-trained models on downstream data is more feasible, as the availability and size of source data are often limited and prohibitive[2]. To maintain control over the factors influencing our study, we focus on *linear probing* pre-trained models, wherein we **freeze the entire model** and train a linear classifier on top of it using the downstream data. Our findings reveal a negative correlation between transfer accuracy and the $\mathcal{NC}_1$ metric on the downstream data. This phenomenon is universal, as it holds true across multiple downstream datasets (Krizhevsky et al., 2009; Maji et al., 2013; Cimpoi et al., 2014; Parkhi et al., 2012), different pre-trained models (He et al., 2016; Dosovitskiy et al., 2021; Huang et al., 2017; Sandler et al., 2018; Radford et al., 2021), the few-shot learning regime, and even across different layers within individual models.

**Pre-trained models with more collapsed last-layer features exhibit better transferability.** To support our claim, we pre-train ResNet50 models on the Cifar-100 dataset using different levels of data augmentations and adversarial training (Madry et al., 2018; Salman et al., 2020; Deng et al., 2021) strength,[3]. Once a model is pre-trained, we evaluate its transfer accuracy on four downstream datasets: Cifar-10 (Krizhevsky et al., 2009), FGVC-Aircraft (Maji et al., 2013), DTD (Cimpoi et al., 2014) and Oxford-IIIT-Pet (Parkhi et al., 2012). As shown in Figure 1, we find a negative (near linear) correlation between $\mathcal{NC}_1$ on Cifar-10 and transfer accuracy on the downstream tasks, where lower $\mathcal{NC}_1$ on Cifar-10 corresponds to higher transfer accuracy.[4] Thus, the $\mathcal{NC}_1$ metric on Cifar-10 can serve as a reliable indicator of transfer accuracy on downstream tasks. To further reinforce our argument, we conduct experiments on the same set of downstream tasks using publicly available pre-trained models on ImageNet-1k (Deng et al., 2009), such as ResNet (He et al., 2016), DenseNet (Huang et al., 2017) and MobileNetV2 (Sandler et al., 2018). In Fig-

---

[2]For instance, the JFT dataset (Sun et al., 2017), used in pretraining the Vision Transformer, is extensive and not publicly accessible.

[3]We use 5 levels of data augmentations, each level represents adding one additional type of augmentation, e.g., Level 1 means Normalization, level 2 means Normalization + RandomCrop, etc. For adversarial training strength, we follow the framework in (Madry et al., 2018) and consider 5 different attack sizes. Please refer to Appendix E for more details.

[4]When evaluating the correlation between $\mathcal{NC}_1$ and transfer accuracy on the same downstream dataset, the correlation is not as strong as we find on Cifar-10, which we discuss in Appendix F.3.

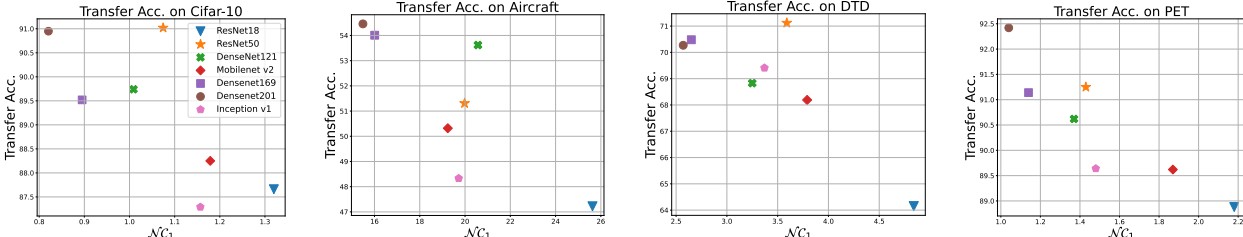

Figure 2: **Transfer accuracy and $\mathcal{NC}_1$ of public ImageNet-1k pre-trained models on different downstream tasks.** We evaluate transfer accuracy and $\mathcal{NC}_1$ on multiple downstream datasets using various ImageNet-1k pre-trained models, such as ResNet (He et al., 2016), DenseNet (Huang et al., 2017) and MobileNetV2 (Sandler et al., 2018). The $\mathcal{NC}_1$ is measured on the corresponding downstream dataset.

Table 2: **Average few-shot classification accuracy and $\mathcal{NC}_1$ metrics on CIFAR-FS and Mini-ImageNet.** The test few-shot accuracies and $\mathcal{NC}_1$ are evaluated on miniImageNet and tieredImageNet meta-test splits. ConvNet4 denotes a 4-layer convolutional network with 64 filters in each layer.

| **Architecture** | ConvNet4 | | ResNet12(He et al., 2016) | | SEResNet12(Hu et al., 2019) | |
|---|---|---|---|---|---|---|
| | 1 shot | 5 shots | 1 shot | 5 shots | 1 shot | 5 shots |
| **Fewshot Accuracy** | | | | | | |
| **CIFAR-FS** | 61.59 | 77.45 | 68.61 | 82.81 | 69.99 | 83.34 |
| **Mini-ImageNet** | 52.04 | 69.07 | 59.52 | 75.92 | 60.21 | 77.17 |
| $\mathcal{NC}_1$ **of models pre-trained on meta-train splits** | | | | | | |
| **CIFAR-FS** | 38.12 | | 29.13 | | 28.40 | |
| **Mini-ImageNet** | 51.82 | | 28.24 | | 25.58 | |

ure 2, we observe the same negative correlation between $\mathcal{NC}_1$ on the downstream data and transfer accuracy, demonstrating that this relationship is not limited to a specific training scenario or network architecture.

Moreover, this negative correlation between downstream $\mathcal{NC}_1$ and transfer accuracy also applies to the few-shot (FS) learning settings, as shown in Table 2 for miniImageNet and CIFAR-FS datasets. Following (Tian et al., 2020), we pre-train different models on the merged meta-training data, and then freeze the models, and learn a linear classifier at meta-testing time. During meta-testing time, support images and query images are transformed into embeddings using the fixed neural network. The linear classifier is trained on the support embeddings. On the query images, we compute the $\mathcal{NC}_1$ of the embeddings and record the few-shot classification accuracies using the linear model.

**Layers with more collapsed output features result in better transferability.** Furthermore, we find the same correlation between the $\mathcal{NC}_1$ metric and transfer accuracy also occurs across different layers of the same pre-trained model. Specifically, as depicted in Figure 5, we linear probe each individual layer of the same pre-trained network, where we use the output of each individual layer as a "feature extractor" and assess its transfer accuracy by training a linear classifier on top of it. Surprisingly, regardless of the layer's depth, if the layer's outputs are more collapsed (smaller $\mathcal{NC}_1$), the resulting features lead to better transfer accuracy. To support our claim, we carried out experiments using the ImageNet-1k pre-trained ResNet34 model (Deng et al., 2009; He et al., 2016). We evaluated the $\mathcal{NC}_1$ metric on each residual block's output feature for the downstream data, as shown in Figure 3 (a). The results indicate that the output features with a smaller $\mathcal{NC}_1$ lead to better transfer accuracy, and there is a near-linear relationship between $\mathcal{NC}_1$ and transfer accuracy, suggesting that transfer accuracy is more closely related to the degree of variability collapse in the layer than the layer's depth. This phenomenon is observed across different network architectures. Figure 3 (b) shows similar results with experiments on the ViT-B (vision transformer base model) (Dosovitskiy et al., 2021) using a pre-trained checkpoint available online.[5] As a result, our findings offer valuable insights into the linear probing of pre-trained models. The conventional practice of solely utilizing last-layer features for linear probing may not be optimal for transfer learning. Instead, employing the $\mathcal{NC}$ metric to identify the optimal layer for linear probing can result in improved performance.

---

[5]The checkpoint used for ViT-B can be found here.

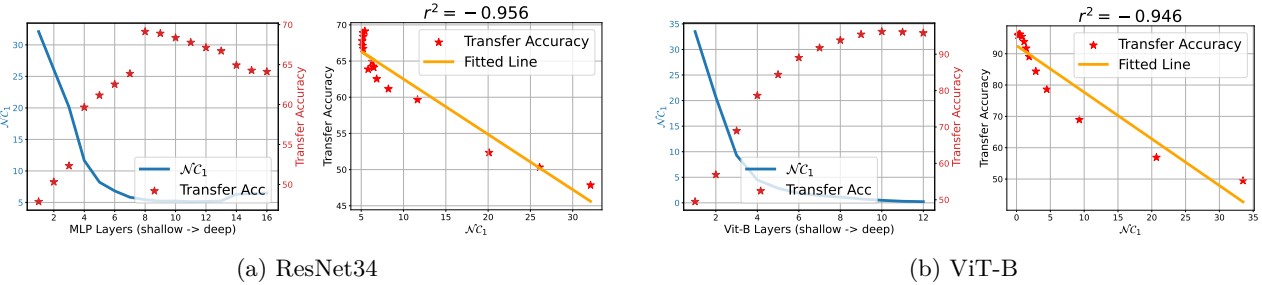

(a) ResNet34         (b) ViT-B

Figure 3: $\mathcal{NC}_1$ **and transfer learning accuracy of different layers from a pre-trained model (Left) and nearly linear relationship between transfer learning accuracy and** $\mathcal{NC}_1$ **(Right).** We use (a) an ImageNet-1k dataset pre-trained ResNet34 model and (b) a released pre-trained ViT-B model. We use the Cifar-10 dataset for transfer learning and measuring the corresponding $\mathcal{NC}_1$.

**Negative correlation between** $\mathcal{NC}_1$ **and downstream accuracy in un-supervised models.** In Figures 1 and 2, we present empirical evidence demonstrating a negative correlation between downstream $\mathcal{NC}$ and downstream transfer accuracy among supervised pre-trained models. To examine our claim in more broad settings, we conducted experiments utilizing CLIP (Dosovitskiy et al., 2021), a model trained on matching image and caption pairs without reliance on labeling information. Specifically, we employed the image encoder of CLIP as a frozen feature extractor to extract features from diverse datasets including Cifar (Krizhevsky et al., 2009), DTD (Cimpoi et al., 2014), FGVC-Aircraft (Maji et al., 2013), Oxford-102-Flower (Nilsback & Zisserman, 2008) and SUN397 (Xiao et al., 2010). Subsequently, we conducted linear probing and computed NC metrics using the extracted features. The transfer accuracy is then plotted against the corresponding $NC_1$ values, as illustrated in Figure 4(a). We can observe from the plot a clear negative correlation between the calculated $\mathcal{NC}_1$ and linear probing accuracy, which extends our claims to un-supervised models.

**Discussion between** $\mathcal{NC}_1$ **and direct classifier measurement.** Despite the empirical evidence shown in this section, an important aspect one may argue is that training a classifier directly on the downstream training data shares the same efficacy with $\mathcal{NC}$ for predicting transfer accuracy. To differentiate between measuring $\mathcal{NC}_1$ and direct classifier measurement, we note that from classical machine learning theory, good training accuracy on the downstream data does not translate to good generalization performance. We might overfit on limited downstream training data but not generalize to the downstream test data. One must test the trained linear classifier on a held-out test set to know a model's actual performance. In contrast, $\mathcal{NC}_1$ can provide valuable insights into a model's transfer performance based solely on the training features from a downstream task. To further demonstrate this point, we conducted an experiment for which we used the image encoder of the CLIP model as a feature extractor to extract training and testing features from multiple downstream datasets. We train linear classifiers evaluate $\mathcal{NC}$ on training features and then calculate training/testing accuracy using the fitted linear classifier. The plots of $\mathcal{NC}_1$ vs transfer accuracy and training accuracy vs. transfer accuracy are shown in Figure 4. We observe that the negative correlation between $\mathcal{NC}$ and transfer accuracy persists even in the case of unsupervised models, while no obvious correlation can be observed for the training accuracy.

## 4 Parameter Efficient Fine-tuning via $\mathcal{NC}$ on Downstream Tasks

Furthermore, our observations, as detailed in the previous section, can provide valuable guidance in designing efficient fine-tuning strategies. In the context of adapting large-scale pre-trained models to downstream vision tasks, there are two common approaches: (i) **linear probing** (Khosla et al., 2020; Kornblith et al., 2021; Deng et al., 2021), which uses the pre-trained model as a feature extractor and only learns a linear classifier for the downstream task, and (ii) **full model fine-tuning** (Dosovitskiy et al., 2021; Kornblith et al., 2019; Salman et al., 2020), which adjusts the entire pre-trained model using downstream training data. Linear probing is a highly parameter-efficient method, while full model fine-tuning, albeit more costly particularly for large-scale foundation models, typically yields superior results. However, in scenarios with limited learning data, full model fine-tuning may lead to overfitting; see Figure 8. To balance these trade-

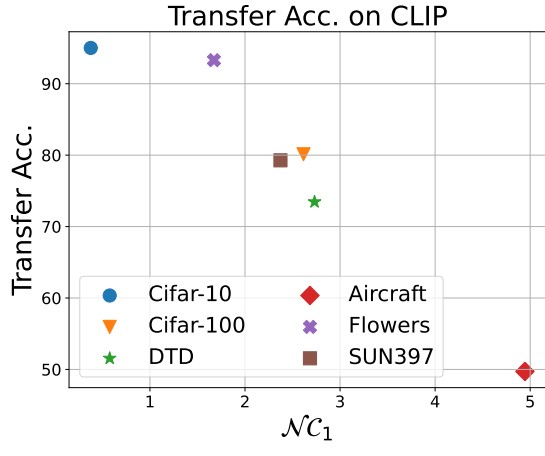 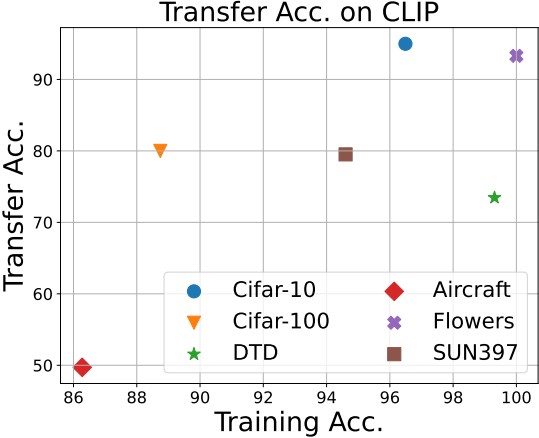

(a) Downstream $\mathcal{NC}_1$ / Trasfer Acc.

(b) Downstream Train Acc. / Trasfer Acc.

Figure 4: **Negative correlation between transfer accuracy and $\mathcal{NC}_1$ of downstream datasets hold on the CLIP model while downstream training accuracy doesn't have a strong correlation with transfer accuracy.** We use the image encoder of CLIP model as a feature extractor to extract training and testing features from multiple downstream datasets. We then train linear classifiers and evaluate $\mathcal{NC}_1$ on training features and evaluate transfer accuracy using the testing features.

offs, recent attention has been given to parameter-efficient fine-tuning in NLP, which selectively adjusts a small portion of network parameters (Hu et al., 2021; Rebuffi et al., 2017; Houlsby et al., 2019). Given the growing trend of foundation models beyond NLP (Dosovitskiy et al., 2021; Radford et al., 2021; Zhai et al., 2021), our work focuses on vision classification tasks, exploring parameter-efficient fine-tuning based on the correlation between last-layer feature variability on downstream data and transfer accuracy. Based upon our extensive experiments in Section 3, we conjecture that

*The optimal transfer accuracy can be achieved by selectively fine-tuning layers to make the last-layer features as collapsed as possible on the downstream training data.*

Based on this, we introduce a simple fine-tuning strategy aimed at increasing feature collapse levels in the last layer, with our results presented in Table 3 and Figure 8. These results show that our proposed method establishs a good balance between linear probing and full model fine-tuning [6], saving over 90% of parameters (see Table 3) and reducing overfitting when training data is scarce (see Figure 8). In the following, we provide a detailed description of our fine-tuning approach.

**Proposed Method: Skip Connection Layer (SCL) Fine-tuning (FT).** Inspired by the observation in Section 3, we propose Skip Connection Layer (SCL) Fine-tuning (FT), which consists of two key components:

- **Fine-tuning one key intermediate layer.** To be parameter efficient, we *only* fine-tune one of the intermediate layers while keeping the rest of the network frozen, which we called it *layer fine-tuning* (Layer FT). To find the optimal layer for fine-tuning, we conduct an ablation study to compare the performances by fine-tuning different intermediate layers with the rest of the network frozen. As shown in Figure 7, fine-tuning the layer closer to the final layer usually leads to more collapsed features[7] in the last layer and, potentially, the better transfer accuracy.This is because the information from the inputs has been better extracted and distilled as getting closer to final layers. On the other hand, in many deep network architectures, layers closer to the output typically have more parameters.[8] Thus, to strike

---

[6]We also compare with MLP probing, for which we replace the final linear classifier by a 2-layer MLP classifier with hidden dimension 1024.

[7]We observe a similar trend for ViT-base models, as shown in Figure 14.

[8]For example, a ResNet18 model has $512 \times 512 \times 3 \times 3$ parameters in the penultimate layer, while only $64 \times 4 \times 3 \times 3$ parameters are in the input layer.

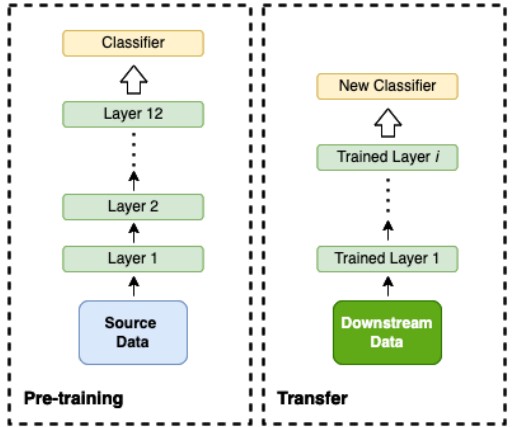

Figure 5: **An illustration of layer-wise transfer learning.** We use a pre-trained model up to the intermediate $i$-th layer as a feature extractor for transfer learning on the downstream tasks.

Figure 6: **Illustrations of Layer FL (left) and SCL FT (right).** Layer FL simply fine-tune one intermediate layer, while SCL FT adds a skip connection to the final layer.

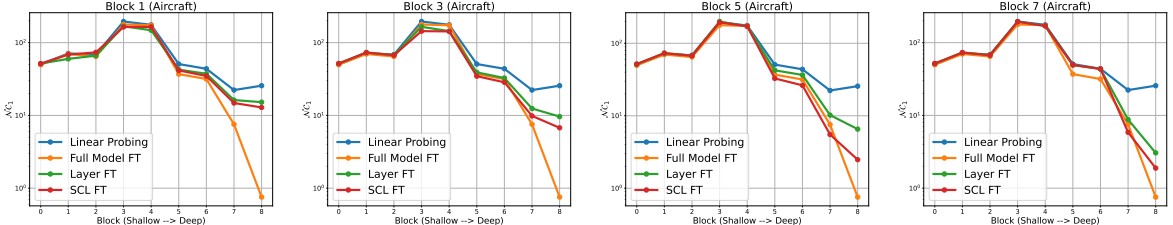

Figure 7: **Layer-wise $\mathcal{NC}_1$ for different fine-tuning methods on ImageNet-1k pre-trained ResNet18.** We compare the layer-wise $\mathcal{NC}_1$ across different fine-tuning methods, including linear probing, Layer FT, and SCL FT, using the ImageNet-1k pre-trained ResNet18 backbone. We evaluate the models on the downstream dataset of FGVC-Aircraft. For the figures from left to right, we plot the results of fine-tuning only Block 1, Block 3, Block 5, and Block 7 in both Layer FT and SCL FT.

a balance between transfer performance and parameter efficiency, we opt to fine-tune one of the middle to deep layers. More specifically, in all our experiments, we fine-tune Block 5 of ResNet18, Block 8 of ResNet50 and Layer 8 for Vit-B32.

- **Improving feature collapse via skip connections.** As illustrated in Figure 6, building on the Layer FT, we introduce a second key component to further the collapse of last-layer feature by adding a skip connection from the key fine-tuned layer to the last layer. We then use the combined features (i.e., the sum of the two outputs) from these layers as the new feature for training the linear classifier and fine-tuning the selected layer. If the dimensions of the features differ (e.g., in CNN-based models), we zero-pad the lower-dimensional feature to match the dimension difference. Our proposed SCL FT method enables more effective fine-tuning of the selected layer by directly passing the data's information from the intermediate layer to the classifier, without losing information through other intermediate layers. Moreover, this approach leverages the depth of deep models, ensuring that the more refined features from the penultimate layer are also passed to the linear classifier. As demonstrated on ResNet in Figure 7, SCL FT leads the most collapsed feature in the last-layer other than full model fine-tuning, and better performance compared with Layer FT. We also observe a similar phenomenon in the case of ViT, for which we postpone to Appendix F.

**Advantages of Our Methods.** Through comprehensive experiments, we demonstrate the superiority of our suggested SCL FT over conventional methods on vision classification problems, such as linear probing, full model fine-tuning, and zero-shot learning. Full model fine-tuning requires retraining all parameters,

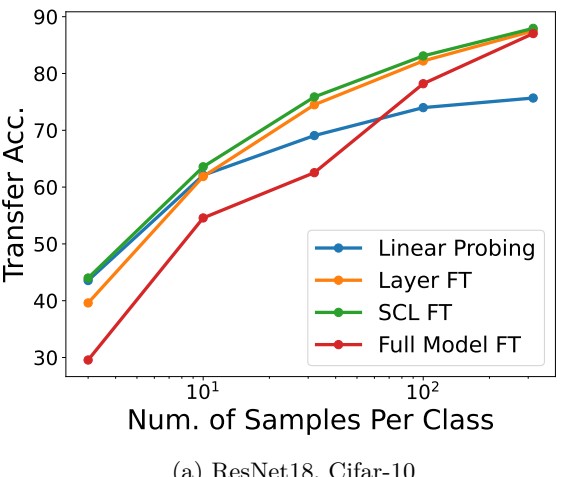 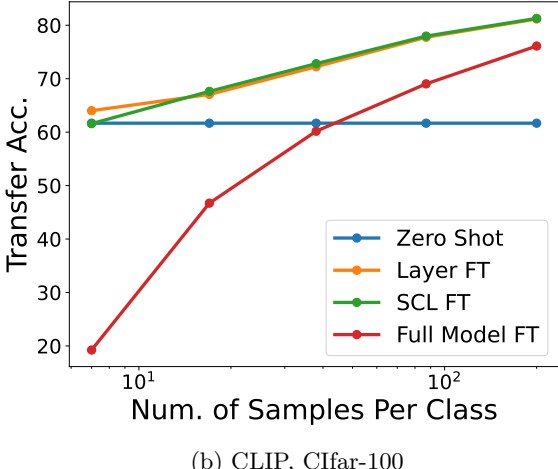

(a) ResNet18, Cifar-10            (b) CLIP, CIfar-100

Figure 8: **Transfer accuracy for different fine-tuning methods with varying size of downstream training dataset.** We fine-tune ImageNet-1k pre-trained ResNet18 models and CLIP using subsets of the Cifar-10 and Cifar-100 downstream datasets, respectively with varying sizes. All results are averaged over 3 runs.)

which amounts to 23 million for ResNet50 or 88 million for ViT-B32. In contrast, our methods, SCL FT and Layer FT, are highly efficient, achieving comparable or better results by fine-tuning only around 8% of the model's parameters, as shown in Table 3. Moreover, our methods exhibit improved resistance to overfitting while achieving equal or superior performance compared to full model fine-tuning; see Figure 8. The detailed experimental setup and hyperparameter details can be found at Appendix E. Below, we discuss two main advantages of our method in detail.

- **Parameter efficiency.** Our method is easily adaptable, working effectively with both CNN-based (ResNets) and transformer-based (ViT) network structures. As demonstrated in Table 3, with just 8% of parameters fine-tuned, SCL FT surpasses linear probing by a great margin and reaches comparable performances compared with full model FT in ViT-based experiments across the datasets FGVC-Aircraft (Maji et al., 2013), DTD (Cimpoi et al., 2014), and Oxford-IIIT-Pet (Parkhi et al., 2012). When applied with ResNet-based backbones, our SCL FT technique still delivers a significant performance boost compared to linear probing in all experiments and reaches full model FT's performance in some scenarios.

- **Mitigating overfitting in the presence of downstream data scarcity.** When dealing with limited downstream data, performing fine-tuning on the entire large-scale pre-trained model can lead to overfitting, resulting in poor generalization performance. In contrast, our methods demonstrate better resilience to data scarcity and decreased overfitting by specifically fine-tuning a small subset of model parameters. To demonstrate this, we conducted fine-tuning experiments on pre-trained ResNet18/CLIP models using varying sizes of subsets from the Cifar-10/Cifar-100 training samples. The outcomes are presented in Figure 8. Our findings reveal that full model fine-tuning is more vulnerable to data scarcity, underperforming linear probing/zero-shot approaches when the downstream training data size is limited. In comparison, our SCL FT and Layer FT methods maintain their robustness and significantly surpass full model fine-tuning until a large amount of downstream training data becomes available.

**Ablation Studies.** There are several design choices to consider when conducting Layer FT. These include decisions on retaining intermediate layers between the fine-tuned layer and the classifier, as well as determining the number of layers to fine-tune. Below, we discuss these two aspects in detail. Additional ablation studies can be found in Appendix F.

- **Effect of removing subsequent layers during Layer FT.** An interesting question regarding Layer Fine-Tuning (Layer FT) is the utility of intermediate layers between the fine-tuned layer and the classifier:

Table 3: **Transfer learning results for linear probing, layer FT, SCL FT and full model FT on downstream datasets.** We use publicly available ResNet50 and ViT-B models. Results reported are averaged over 3 random seeds. We use bold font to represent the best result for each dataset and underline the second-best result.

| Backbone | ResNet50 | | | | | ViT-B32 | | |
|---|---|---|---|---|---|---|---|---|
| Dataset | Cifar-10 | Cifar-100 | Aircraft | DTD | PET | Aircraft | DTD | PET |
| **Transfer accuracy** | | | | | | | | |
| Linear Probe | $85.41 \pm 0.01$ | $65.38 \pm 0.05$ | $44.83 \pm 0.14$ | $68.87 \pm 0.29$ | $90.89 \pm 0.09$ | $50.74 \pm 0.21$ | $74.92 \pm 0.15$ | $92.67 \pm 0.17$ |
| MLP Probe | $85.60 \pm 0.13$ | $65.94 \pm 0.05$ | $45.54 \pm 0.15$ | $70.12 \pm 0.03$ | $90.78 \pm 0.03$ | $51.43 \pm 0.37$ | $75.39 \pm 0.21$ | $93.71 \pm 0.25$ |
| Layer FT | $94.60 \pm 0.11$ | $\underline{79.13 \pm 0.09}$ | $71.98 \pm 0.49$ | $72.62 \pm 0.52$ | $\underline{91.95 \pm 0.10}$ | $\underline{73.20 \pm 0.47}$ | $\underline{77.51 \pm 0.18}$ | $93.28 \pm 0.08$ |
| SCL FT | $\underline{95.08 \pm 0.14}$ | $79.04 \pm 0.19$ | $72.57 \pm 0.14$ | $\underline{73.51 \pm 0.04}$ | $\mathbf{92.07 \pm 0.02}$ | $73.17 \pm 0.37$ | $\mathbf{77.80 \pm 0.17}$ | $\underline{93.46 \pm 0.05}$ |
| Full Model FT | $\mathbf{95.25 \pm 0.07}$ | $\mathbf{79.57 \pm 0.13}$ | $\mathbf{83.52 \pm 0.12}$ | $\mathbf{75.20 \pm 0.10}$ | $86.25 \pm 0.06$ | $\mathbf{74.58 \pm 0.35}$ | $77.41 \pm 0.59$ | $\mathbf{93.51 \pm 0.10}$ |
| $\mathcal{NC}_1$ **evaluated on the penultimate layer feature** $h^{L-1}$ | | | | | | | | |
| Linear Probe | 1.81 | 18.40 | 19.84 | 3.54 | 1.49 | 17.90 | 1.99 | 0.66 |
| MLP Probe | 1.81 | 18.40 | 19.84 | 3.54 | 1.49 | 17.90 | 1.99 | 0.66 |
| Layer FT | $0.29 \pm 0.03$ | $3.44 \pm 0.14$ | $2.71 \pm 0.03$ | $1.14 \pm 0.06$ | $0.56 \pm 0.04$ | $4.75 \pm 0.00$ | $1.02 \pm 0.09$ | $0.33 \pm 0.03$ |
| SCL FT | $0.19 \pm 0.02$ | $2.40 \pm 0.15$ | $1.23 \pm 0.19$ | $0.65 \pm 0.06$ | $0.39 \pm 0.01$ | $4.97 \pm 0.03$ | $1.13 \pm 0.06$ | $0.37 \pm 0.01$ |
| Full Model FT | $0.04 \pm 0.00$ | $0.06 \pm 0.01$ | $0.25 \pm 0.02$ | $0.25 \pm 0.02$ | $0.12 \pm 0.00$ | $3.33 \pm 0.21$ | $0.97 \pm 0.05$ | $0.14 \pm 0.00$ |
| **Percentage of parameters fine-tuned** | | | | | | | | |
| Linear Probe | 0.09% | 0.86% | 0.86% | 0.41% | 0.32% | 0.09% | 0.04% | 0.03% |
| MLP Probe | 8.23% | 8.56% | 8.56% | 8.37% | 8.33% | 1.00% | 0.94% | 0.93% |
| Layer FT | 6.52% | 7.24% | 7.24% | 6.82% | 6.73% | 8.18% | 8.14% | 8.13% |
| SCL FT | 6.52% | 7.24% | 7.24% | 6.82% | 6.73% | 8.18% | 8.14% | 8.13% |
| Full Model FT | 100% | 100% | 100% | 100% | 100% | 100% | 100% | 100% |

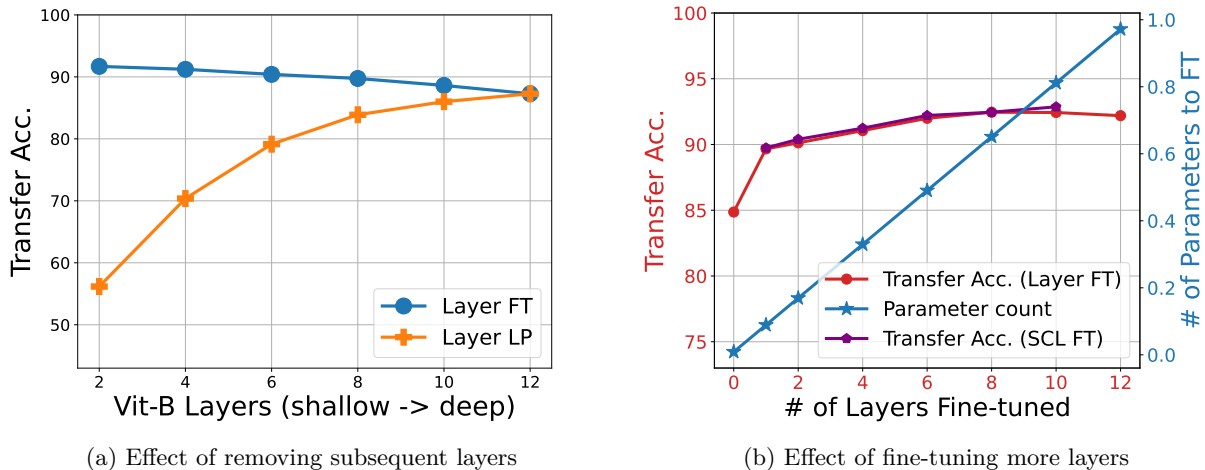

(a) Effect of removing subsequent layers      (b) Effect of fine-tuning more layers

Figure 9: We fine-tune the ViT model on the Cifar-100 dataset. **(a): Transfer accuracy drops when removing subsequent layers during Layer FT.** We conduct Layer FT and Layer LP (removing subsequent layers on top of the fine-tuned layer); **(b): Fine-tuning more layers brings mediocre performance boost while using significantly more parameters.** 0 in the x-axis denotes linear probing and 12 denotes full model fine-tuning.

can these layers be discarded without negatively impacting performance? To investigate this, we conducted a series of experiments with the ViT model on the CIFAR-100 dataset, employing both Layer FT and Layer LP approaches. During fine-tuning, Layer LP removes all layers subsequent to the fine-tuned layer and directly utilizes its features for classification. The accuracy results for both methods are detailed in Figure 9(a). We can observe that the elimination of subsequent layers significantly compromises performance in comparison to Layer FT, with the degree of performance decline being positively correlated to the number of layers removed. This finding indicates that the layers following the fine-tuned one, even

though not changed during fine-tuning, play a crucial role in deriving semantically rich information for classification.

- **Effect of fine-tuning more layers.** We also explore the impact of unfreezing more layers during model fine-tuning. Specifically, we fine-tune the ViT-B32 model on the CIFAR-100 dataset, varying the number of layers that are fine-tuned. As illustrated in Figure 9(b), the parameter increase is linear as we unfreeze more layers, yet the improvement in performance is relatively marginal when compared to the gains observed from linear probing to Layer FT. Therefore, to strike a balance between parameter efficiency and transfer learning performance, we focus on only fine-tuning one layer of the entire model in this work.

## 5 Limitations of pre-training $\mathcal{NC}$ in predicting transfer accuracy

Numerous studies have explored the link between feature diversity and transfer accuracy from a variety of angles (Islam et al., 2021; Kornblith et al., 2021; Feng et al., 2021; Galanti et al., 2022b). Additionally, the rise of large-scale foundational models (OpenAI, 2023; Radford et al., 2021; Kirillov et al., 2023), coupled with the inaccessibility of much of the pre-trained data for these models, makes the evaluation of $\mathcal{NC}$ on the source dataset impractical with these pre-trained models. Therefore, we focus on establishing the relationship between downstream $\mathcal{NC}$ and transferability in this work. To complement our study, we have done some small-scale experiments to study the relationship between the $\mathcal{NC}$ metrics and transfer accuracy on the source training dataset (I.e., we measure $\mathcal{NC}$ on the source training dataset). These experiments revealed a positive correlation between source $\mathcal{NC}$ and transferability, but only to a certain extent which prevents us from reaching a general conclusion regarding this relationship. Therefore, we leave these results and discussions to Appendix B and include only a representative example in this section to illustrate that relying on pre-training NC to predict transfer accuracy may not be as reliable as utilizing downstream NC, as demonstrated in earlier sections.

To demonstrate the limitation of source $\mathcal{NC}$ , we pre-train ResNet50 models on the Cifar-100 dataset using different levels of data augmentations and adversarial training (Madry et al., 2018; Salman et al., 2020; Deng et al., 2021) strength,[9] and then report the transfer accuracy of the pre-trained models on the Cifar-10 dataset. As shown in Figure 10, we observe that there exists a positive correlation between the $\mathcal{NC}_1$ on the source dataset and the transfer accuracy but the correlation only holds up to a certain threshold. We found it is generally hard to locate the turning point where the positive correlation diminishes across different experimental setups. In Figure 10, this turning point appears to be around the value of 1.0. However, in a distinct experimental scenario, as depicted in Figure 12, the positive correlation persists even when the pre-trained $\mathcal{NC}$ exceeds 2.0.

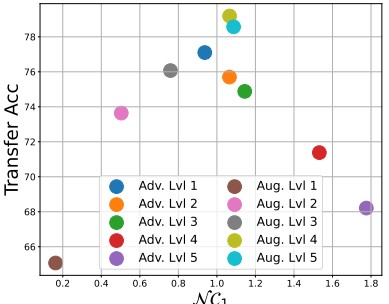

Figure 10: $\mathcal{NC}_1$ **vs. transfer learning accuracy.** Models are pre-trained on the Cifar-100 dataset with different levels of data augmentation and adversarial training strength, and then transfer accuracy of pre-trained models is reported on the Cifar-10 dataset.

We believe the reason behind the limitation of source $\mathcal{NC}$ is that the magnitude of $\mathcal{NC}_1$ is affected by two factors of the learned features: ($i$) within class feature diversity and ($ii$) between class discrimination. When the source $\mathcal{NC}_1$ becomes too large, the features lose the between-class discrimination, which results in poor transfer accuracy. An extreme case would be an untrained deep model with randomly initialized weights. Obviously, it possesses large $\mathcal{NC}_1$ with large feature diversity, but its features have poor between-class discrimination. Therefore, random features have poor transferability. To better predict the model transferability, we need more precise metrics for measuring both the within-class feature diversity and between-class discrimination, where the two could have a tradeoff between each other. We leave the investigation as future work.

---

[9]We use 5 levels of data augmentations, each level represents adding one additional type of augmentation, e.g., Level 1 means Normalization, level 2 means Normalization + RandomCrop, etc. For adversarial training strength, we follow the framework in (Madry et al., 2018) and consider 5 different attack sizes. Please refer to Appendix E for more details.

# 6 Conclusion

In this work, we have explored the relationship between the degree of feature collapse, as measured by the $\mathcal{NC}$, and transferability in transfer learning. Our findings show that there is a twofold relationship between $\mathcal{NC}$ and transferability: ($i$) more collapsed features on the downstream data leads to better transfer performance; and ($ii$) models that are less collapsed on the source data have better transferability up to a certain threshold. This relationship holds both across and within models. Based on these findings, we propose a simple yet effective model fine-tuning method with significantly reduced number of fine-tuning parameters. Our experiments show that our proposed method can achieve comparable performance compared to full model FT across various tasks and setups. Further discussions of future directions are deferred to Appendix C.

## Acknowledgement

XL and QQ are grateful for the generous support from NSF CAREER CCF 2143904, NSF CCF 2212066, NSF CCF 2212326, NSF IIS 2312842, ONR N00014-22-1-2529, and an AWS AI Award. JZ and ZZ express their appreciation for the support received from NSF grants CCF-2240708 and CCF-2241298. SL acknowledges support from NSF NRT-HDR award 1922658 and Alzheimer's Association grant AARG-NTF-21-848627, while CFG acknowledges support from NSF OAC-2103936. We would also like to thank Dr. Chong You (Google Research) and Dr. Hongyuan Mei (TTIC) for their valuable discussions and support throughout this work.

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

# Appendix

The appendices are organized as follows. In Appendix A, we discuss related recent advances in neural collapse, model pre-training and fine-tuning. In Appendix B, we provide more experiments regarding source $\mathcal{NC}$ and transferability. In Appendix C, we discuss the future directions inspired by this work that we want to pursue. In Appendix D, we review other metrics that measure feature diversity and discrimination. In Appendix E, we provide experimental details for each figure and each table in the main body of the paper. Finally, in Appendix F, we provide complimentary experimental results for Section 3, and the proposed fine-tuning methods in Section 4.

## A  Related work

Our work can be related to many existing results on neural collapse, model pre-training, and parameter-efficient fine-tuning. Below, we provide a summary and brief discussion of these results.

- **Studies of the Neural Collapse Phenomenon.** The occurrence of the $\mathcal{NC}$ phenomenon has recently gained significant attention in both practical and theoretical fields (Papyan et al., 2020; lu2, 2022; Zhu et al., 2021; Fang et al., 2021; Han et al., 2022; Kothapalli, 2023), with studies exploring its implications on training, generalization, and transferability of deep networks; see a recent review paper (Kothapalli, 2023). Most of these existing studies focused on training. For instance, various loss functions, such as cross-entropy (Papyan et al., 2020; lu2, 2022; Zhu et al., 2021; Fang et al., 2021; Ji et al., 2022; Yaras et al., 2022; Liu et al., 2020), mean-squared error (Mixon et al., 2020; Han et al., 2022; Zhou et al., 2022; Tirer & Bruna, 2022; Rangamani & Banburski-Fahey, 2022; Wang et al., 2022; Dang et al., 2023), and supervised contrastive (Graf et al., 2021) losses, have been shown to exhibit $\mathcal{NC}$ during training. Other lines of work have investigated the relationship between imbalanced training and $\mathcal{NC}$ (Fang et al., 2021; Xie et al., 2022; Yang et al., 2022; Thrampoulidis et al., 2022; Behnia et al., 2022; Zhong et al., 2023; Sharma et al., 2023; Behnia et al., 2023), generalization to the settings of large classes (Liu et al., 2023c; Gao et al., 2023; Jiang et al., 2023). Furthermore, a line of work found the phenomenon of progressive feature variability collapse across network layers on the pre-training dataset (Hui et al., 2022; Papyan, 2020; He & Su, 2022; Rangamani et al., 2023; Wang et al., 2023a)[10]. Recent works have also investigated the relationships between $\mathcal{NC}$ and both generalization and transferability (Hui et al., 2022; Galanti et al., 2022b;a; Galanti, 2022; Chen et al., 2022a; Wang et al., 2023b). In particular, the study in (Galanti et al., 2022b;a) reveal that $\mathcal{NC}$ occurs on test data when drawn from the same distribution as training data, but the collapse is less severe for finite test samples (Hui et al., 2022). While Galanti et al. (2022b;a) also investigated the correlation between $\mathcal{NC}$ and transfer learning, their study focused solely on the influence of the number of source classes. Our research, on the other hand, is much more comprehensive and demonstrates a universal correlation between $\mathcal{NC}$ and transfer accuracy on downstream data [11]. We examine a broad range of settings, including various tasks, pre-trained model architectures, training setups, and even different layers within individual models. Furthermore, our investigation into $\mathcal{NC}$ has contributed to the principled development of more efficient fine-tuning methods.

- **Model Pre-training & Fine-Tuning.** While various studies have explored factors influencing transferability during model pre-training (Kornblith et al., 2019; 2021; Nayman et al., 2022), their findings largely remain ambiguous. Additionally, these studies primarily focus on characterizing transferability based on the source dataset, which is often inaccessible in the era of large-scale pre-trained foundation models (Dosovitskiy et al., 2021; Radford et al., 2021; OpenAI, 2023; Liu et al., 2023a; Touvron et al.,

---

[10]Progressive variability collapse happens for well-trained models when feature diversity is evaluated on the source dataset. Due to distribution shift, this pattern is not exact when we evaluating feature diversity on the downstream datasets, see Figure 7 for an example.

[11]The work (Wang et al., 2023b) also attempt to establish connections between $\mathcal{NC}$ on downstream data and transfer accuracy. However, their metric is more intricate than ours. Moreover, their study is exclusively concentrated on $\mathcal{NC}$ in downstream tasks of various pre-trained models. In contrast, our research encompasses a more comprehensive scope, examining both inter-model and intra-model $\mathcal{NC}$ across diverse training configurations.

2023). In contrast, our work primarily concentrates on evaluating the transferability of pre-trained models using downstream data. This approach not only enables us to establish a universal relationship between $\mathcal{NC}$ and transferability, but also aligns with the current landscape of large-scale pre-trained models that have limited access to source data. In the field of vision model fine-tuning, two commonly used approaches are linear probing and full model fine-tuning (Kumar et al., 2022). Linear probing focuses on training only the classifier, resulting in parameter-efficient models that often underperform compared to full model fine-tuning. Previous studies have attempted to strike a balance between parameter efficiency and performance by incorporating "adaptors" as auxiliary subnetworks between pre-trained layers (Rebuffi et al., 2017; Houlsby et al., 2019). While effective for transformers in NLP and recently extended to multi-modality and vision domains (Gao et al., 2021; Zhang et al., 2021; Chen et al., 2022b), this approach requires additional structures and is specific to certain network architectures. In contrast, our fine-tuning method does not introduce extra components and can be easily applied to any network architecture. Another line of work involves storing all intermediate layer features and linear probing a subset of them (Evci et al., 2022; Adler et al., 2020). While this approach shows promise when training data for downstream tasks is scarce, it fails to compete with full model fine-tuning when ample training data is available. In contrast, our approach is applicable regardless of data volume and is more memory-efficient since it does not require storing all intermediate features. Furthermore, certain methods aim to adapt pre-trained models to downstream tasks by updating low-rank counterparts of weight matrices (Hu et al., 2021; He et al., 2022; Zhang et al., 2023; Chavan et al., 2023). These methods often involve modifying multiple components of pre-trained models at different layers, whereas our approach focuses on updating one single layer of the overall model.

## B  Additional observations on source $\mathcal{NC}$ and transferability

In Section 5, we have demonstrated the limitations of source $\mathcal{NC}$ in predicting transferability. In this section, we discuss this relationship in more detail by showing that the choice of training losses and the design of network architecture have a direct impact on the levels of feature collapse on the penultimate layer and the transfer accuracy. Similar observations were made in (Islam et al., 2021; Kornblith et al., 2021) regarding the impact of different training losses on transfer performance. To demonstrate this, we pre-trained ResNet18 models on the Cifar-100 dataset using three different loss functions: CE, MSE (Hui & Belkin, 2020), and SupCon (Khosla et al., 2020). We then evaluated the test accuracy on the Cifar-10 dataset by training a linear classifier on the frozen pre-trained models. For the SupCon loss, we followed the setup described in (Khosla et al., 2020), which uses an MLP as a projection head after the ResNet18 encoder. After pre-training, we only used the encoder network as the pre-trained model for downstream tasks and abandoned the projection head. Based on the experiment results, we observe the following.

- *The choice of training loss impacts feature diversity, which in turn affects transfer accuracy.* Larger feature diversity, as measured by a larger $\mathcal{NC}_1$ value, generally leads to better transfer accuracy. As demonstrated in Appendix C, where the last two columns reports the results for SupCon with different layers of projection heads, a model pre-trained with the MSE loss exhibits a severely collapsed representation on the source dataset, with the smallest $\mathcal{NC}_1$ value and the worst transfer accuracy. While SupCon with projection head results in higher $\mathcal{NC}_1$, and better transfer accuracy.

- *The MLP projection head is crucial for improved transferability.* The model pre-trained with the SupCon loss and a multi-layer MLP projection head shows the least feature collapse compared to the other models and demonstrates superior transfer accuracy.[12] If we substitute the MLP with a linear projection layer, both the $\mathcal{NC}_1$ metric and transfer accuracy of SupCon decrease, resulting in performance comparable to the models pre-trained with the CE loss. This can also be demonstrated by our experiments in Figure 12, where we pre-train ResNet-50 models on the Cifar-100 dataset and report $\mathcal{NC}$ metrics and transfer accuracy for varying numbers of projection head layers (from one to three layers). Our results show that using projection heads significantly increases representation diversity and transfer accuracy – adding more layers of MLP projection leads to higher $\mathcal{NC}_1$ and improved transfer accuracy, although the performance improvement quickly plateaus at three layers of MLP. This could suggest that the effectiveness of projec-

---

[12]This observation aligns with recent work (Islam et al., 2021).

| Training | $\parallel$ MSE (w/o proj.) | Cross-entropy (w/o proj.) | SupCon (w/ linear proj.) | SupCon (w/ mlp proj.) |
|---|---|---|---|---|
| $\mathcal{NC}_1$ **(Cifar-100)** | 0.001 | 0.771 | 0.792 | 2.991 |
| **Transfer Acc.** | 53.96 | 71.2 | 69.89 | 79.51 |

tion heads in model pre-training is not limited to contrastive losses but applies universally across various training loss types (e.g. CE and MSE).

Although we found the positive correlation between source $\mathcal{NC}$ and transferability exists in certain scenarios, our discussion in Section 5 demonstrates that there exists a certain threshold where the positive correlation vanishes. This phenomenon could be attributed to a tradeoff between feature diversity and class separation, which will be a compelling topic to study in future work. Finally, We note that since our experiments in this section are conducted on small-scale datasets, the observations may not hold in general.

## C   Additional Discussions

Besides examining the relationship between $\mathcal{NC}$ and transfer learning regarding both source and target data, our findings also open up new avenues for future research that we discuss in the following.

**Neuron collapse is not a pure training phenomenon.**   Previous work (Hui et al., 2022) points out that $\mathcal{NC}$ is mainly an optimization phenomenon that does not necessarily relate to generalization (or transferability). Our work, on one hand, corroborates with the finding that pretraining $\mathcal{NC}$ does not always suggest better transferability, but also shows a positive correlation between pretraining $\mathcal{NC}$ and transferability to a certain extent. On the other hand, our work also shows that downstream $\mathcal{NC}$ on a dataset where $\mathcal{NC}$ is well-defined correlates with the transfer performances across different datasets and thus could be a general indicator for the transferability. This suggests that $\mathcal{NC}$ may not be merely an optimization phenomenon. An important future direction we will pursue is to theoretically understand the connection between transferability and $\mathcal{NC}$.

**Boost model transferability by insights from $\mathcal{NC}$.**   Our findings can be leveraged to enhance model transferability from two perspectives. Firstly, the correlation between pretraining $\mathcal{NC}$ and transferability suggests that increasing $\mathcal{NC}_1$ to a certain extent can boost transferability, which can be accomplished through popular methods like multi-layer projection heads and data augmentation. We expect that other methods that focus on the geometry of the representations could also be developed. Secondly, by showing the strong correlation between downstream $\mathcal{NC}$ and transfer accuracy, we can devise simple but effective strategies for efficient transfer learning. Although our simple approach may not be the optimal way to take advantage of this relationship, we believe that there are more potent methods that could exploit this relationship better and therefore lead to better transferability. We leave this as a topic for future investigation.

**Transfer learning beyond $\mathcal{NC}$.**   Our results indicate that $\mathcal{NC}$ on the source datasets is related to transferability to a certain extent. However, identifying the threshold and explaining the change in transferability beyond this threshold is challenging using the $\mathcal{NC}$ framework alone. Moreover, solely relying on $\mathcal{NC}$ to study transfer learning may not always be appropriate. For instance, recent research (Wang & Isola, 2020) demonstrated that representations learned through unsupervised contrastive methods are uniform across hyperspheres, and (Chan et al., 2021) showed that representations learned using the principle of maximal coding rate reduction (MCR$^2$) form subspaces rather than collapse to single points. Therefore, further disclosing the mysteries shrouded around transfer learning would require new frameworks and new tools, which we leave for future investigation.

## D   Other Metrics for Measuring $\mathcal{NC}$

**Numerical rank of the features $\boldsymbol{H}$.**   The $\mathcal{NC}_1$ does not directly reveal the dimensionality of the features spanned for each class. In this case, measuring the rank of the features ($\boldsymbol{H}_k$) is more suitable. However,

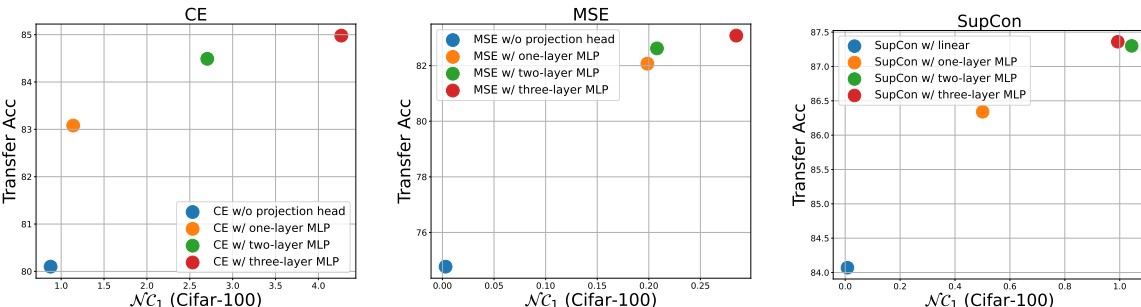

Figure 11: **Trend of $\mathcal{NC}_1$ during training and transfer learning accuracy of the pretrained models.** ResNet-50 models are pretrained using Cifar-100 dataset with CE loss (Left), MSE loss (Middle) and SupCon loss (Right). Models are pretrained with different numbers of layers for projection heads and transfered on the Cifar-10 dataset.

Figure 12: **Comparisons of $\mathcal{NC}_1$ and transfer accuracy with different training losses and settings.** We pre-train ResNet18 models on the Cifar-100 dataset, and then test the models on Cifar-10. We use proj. to denote projection head; w/o proj. means without projection head, w/linear proj. means adding one linear layer projection layer, and w/ mlp proj. means adding a two-layer MLP projection.

the calculations for both $\mathcal{NC}_1$ and rank are expensive when feature dimension gets too large. Thus, we emphasize introducing *numerical rank* (Timor et al., 2022) as an approximation.

$$\widetilde{\mathrm{rank}}(\boldsymbol{H}) \; := \; \frac{1}{K} \sum_{k=1}^{K} \|\boldsymbol{H}_k\|_F^2 \, / \, \|\boldsymbol{H}_k\|_2^2 \,,$$

where $\|\cdot\|_F$ represents the Frobenius norm and $\|\cdot\|_2$ represents the Spectral norm. $\widetilde{\mathrm{rank}}(\boldsymbol{H})$(*numerical rank*) could be seen as an estimation of the true rank for any matrix. Note that we use the Power Method to approximate the Spectral norm. The metric is evaluated by averaging over all classes, and a smaller $\widetilde{\mathrm{rank}}(\boldsymbol{H})$ indicates more feature collapse towards their respective class means.

**Class-distance normalized variance (CDNV) (Galanti et al., 2022b).** To alleviate the computational issue of $\mathcal{NC}_1$, the *class-distance normalized variance* (CDNV) introduced in (Galanti et al., 2022b) provides an alternative metric that is inexpensive to evaluate. Let $\mathcal{X}$ denote the space of the input data $\boldsymbol{x}$, and let $\boldsymbol{Q}_k$ be the distribution over $\mathcal{X}$ conditioned on class $k$. For two different classes with $\boldsymbol{Q}_i$ and $\boldsymbol{Q}_j$ $(i \neq j)$, the CDNV metric can be described by the following equation:

$$V_{\phi_{\boldsymbol{\theta}}}(\boldsymbol{Q}_i, \boldsymbol{Q}_j) = \frac{\mathrm{Var}_{\phi_{\boldsymbol{\theta}}}(\boldsymbol{Q}_i) + \mathrm{Var}_{\phi_{\boldsymbol{\theta}}}(\boldsymbol{Q}_j)}{2\|\mu_{\phi_{\boldsymbol{\theta}}}(\boldsymbol{Q}_i) - \mu_{\phi_{\boldsymbol{\theta}}}(\boldsymbol{Q}_j)\|_2^2},$$

where $\mu_{\phi_{\boldsymbol{\theta}}}(\boldsymbol{Q}_k) = \mathbb{E}_{\boldsymbol{x} \sim \boldsymbol{Q}_k}[\phi_{\boldsymbol{\theta}}(\boldsymbol{x})]$ denotes the class-conditional feature mean and $\mathrm{Var}_{\phi_{\boldsymbol{\theta}}}(\boldsymbol{Q}_k) = \mathbb{E}_{\boldsymbol{x} \sim \boldsymbol{Q}_k}[\|\phi_{\boldsymbol{\theta}}(\boldsymbol{x}) - \mu_{\phi_{\boldsymbol{\theta}}}(\boldsymbol{Q}_k)\|^2]$ denotes the feature variance for the distribution $\boldsymbol{Q}_k$. Although the exact expectation is impossible to evaluate, we can approximate it via its empirical mean and empirical variance on the given training samples as

$$\widehat{V}_{\phi_{\boldsymbol{\theta}}}(\boldsymbol{Q}_i, \boldsymbol{Q}_j) \;=\; \frac{\widehat{\mathrm{Var}}_{\phi_{\boldsymbol{\theta}}}(\boldsymbol{Q}_i) + \widehat{\mathrm{Var}}_{\phi_{\boldsymbol{\theta}}}(\boldsymbol{Q}_j)}{2\|\widehat{\mu}_{\phi_{\boldsymbol{\theta}}}(\boldsymbol{Q}_i) - \widehat{\mu}_{\phi_{\boldsymbol{\theta}}}(\boldsymbol{Q}_j)\|^2},$$

$$\widehat{\mu}_{\phi_{\boldsymbol{\theta}}}(\boldsymbol{Q}_k) \;=\; \frac{1}{n_k} \sum_{i=1}^{n_k} \phi_{\boldsymbol{\theta}}(\boldsymbol{x}_{k,i}), \; \widehat{\mathrm{Var}}_{\phi_{\boldsymbol{\theta}}}(\boldsymbol{Q}_k) \;=\; \frac{1}{n_k} \sum_{i=1}^{n_k} \|\phi_{\boldsymbol{\theta}}(\boldsymbol{x}_{k,i}) - \mu_{\phi_{\boldsymbol{\theta}}}(\boldsymbol{X}_k)\|^2.$$

To characterize the overall degree of collapse for a model, we can use the empirical CDNV between all pairwise classes (i.e., $\mathrm{Avg}_{i \neq j}[\widehat{V}_{\phi_{\boldsymbol{\theta}}}(\boldsymbol{Q}_i, \boldsymbol{Q}_j)]$). If a model achieves perfect collapse on the data, we have $\mathrm{Avg}_{i \neq j}[\widehat{V}_{\phi_{\boldsymbol{\theta}}}(\boldsymbol{Q}_i, \boldsymbol{Q}_j)] = 0$. Because the CDNV metric is purely norm-based, computational complexity scales linearly with the feature dimension $d$, making it a good surrogate for $\mathcal{NC}_1$ when the feature dimension $d$ is large.

# E   Extra Experimental Details

In this section, we provide additional technical details for all the experiments in the main body of the paper to facilitate reproducibility. In particular, Appendix E.1 includes all the experimental details for the figures (from Figure 1 to Figure 8), and Appendix E.2 includes all the experimental details for the tables (Table 2 and Table 3).

**General experiment setups.** We perform all experiments using a single NVIDIA A40 GPU, most experiments could be finished in less than 4 hours. Unless otherwise specified, all pre-training and transfer learning are run for 200 epochs using SGD with a momentum of 0.9, a weight decay of $1 \times 10^{-4}$, and a dynamically changing learning rate ranging from $1 \times 10^{-1}$ to $1 \times 10^{-4}$ controlled by a CosineAnnealing learning rate scheduler as described in (Loshchilov & Hutter, 2017). When using ImageNet pre-trained models, we resize each input image to $224 \times 224$ for training, testing, and evaluating $\mathcal{NC}$.

## E.1   Technical Detail for the Figures

**Experimental details for Figure 1 and Figure 2.** In Figure 1, we pre-train ResNet50 models using different levels of data augmentation and adversarial training. For data augmentation, we consider RandomCrop, RandomHorizontalFlip, ColorJitter, and RandomGrayScale. We increase the levels of data augmentation by adding one additional type of augmentation to the previous level. For Level 1, we only do the standardization for samples to make the values have mean 0, variance 1; for level 2, we add Random-Crop to the Level 1 augmentation; for level 3, we add RandomHorizontalFlip to the previous level, and so on, so forth. Previous research has demonstrated the effectiveness of adversarial pre-training for enhancing transferability (Salman et al., 2020; Deng et al., 2021). To investigate the impact of adversarial training with more detail, we adopted the $\ell_\infty$-norm bounded adversarial training framework proposed in (Madry et al., 2018), employing five different levels of attack sizes: $\frac{1}{255}$, $\frac{2}{255}$, $\frac{3}{255}$, $\frac{5}{255}$, and $\frac{8}{255}$. Our experimental results, illustrated in Figure 1, confirm that using smaller attack sizes in adversarial training can enhance the transferability of pre-trained models, while larger step sizes can negatively impact the transferability. We transfer the pre-trained models to four different downstream datasets: Cifar-10, FGVC-Aircraft, DTD, and Oxford-IIIT-Pet. Although there are many other benchmark datasets to use, we choose these four because the number of samples for each class is balanced for the ease of study. This aligns with the scenario where $\mathcal{NC}$ is studied in (Papyan et al., 2020). In Figure 1, we pre-train ResNet50 models on Cifar-100 dataset, while in Figure 2 we conduct similar experiments using public avaiable pre-trained models on ImageNet-1k (Deng et al., 2009) including ResNet (He et al., 2016), DenseNet (Huang et al., 2017), and Mobilenet-v2 (Sandler et al., 2018)

**Experimental details for Figure 3.** In Figure 3 (a), we used a pre-trained ResNet34 (He et al., 2016) model trained on ImageNet-1k (Deng et al., 2009). To simplify the computation of $\mathcal{NC}_1$, we extracted the output from each residual block as features (as demonstrated in Figure 13), and then conducted adaptive average pooling on these features, ensuring that each channel contained only one element. We then computed $\mathcal{NC}_1$ and performed transfer learning on the resulting features. In Figure 3 (b), we used a pre-trained Vit-B32 (Dosovitskiy et al., 2021) model that is publicly available online. For each encoder layer $l$ in Vit-B32, the output features $\boldsymbol{h}_l$ are of dimension 145 (which includes the number of patches plus an additional classification token) $\times$ 768 (the hidden dimension), i.e., $\boldsymbol{h}_l \in \mathbb{R}^{145 \times 768}$. To conduct our layer-wise transfer learning experiment, we first applied average pooling to the 145 patches to reduce the dimensionality of the feature vectors. Specifically, we used an average pooling operator $ap : \mathbb{R}^{145 \times 768} \mapsto \mathbb{R}^{768}$ to downsample $\boldsymbol{h}_l$, resulting in $\widehat{\boldsymbol{h}}_l = ap(\boldsymbol{h}_l)$. Subsequently, we trained a linear classifier using the downsampled features $\widehat{\boldsymbol{h}}_l$ on top of each encoder layer.

**Experimental details for Figure 7.** In Figure 7, we investigate the effects of different fine-tuning methods on the layerwise $\mathcal{NC}_1$ when we are fine-tuning ResNet18 models pre-trained on ImageNet-1k. To facilitate

the computation of $\mathcal{NC}_1$, we adopt the same adaptive pooling strategy as in Figure 3 on the features from each residual block of the ResNet model.[13]

**Experimental details for Figure 8.** We empirically demonstrate that our proposed methods are more robust to data scarcity compared with linear probing and full model FT. To this end, we transfer the ImageNet-1k pre-trained ResNet18 model on subsets of Cifar-10 with varying sizes. More specifically, we select the sizes from a logarithmically spaced list of values: [3, 10, 32, 100, 316] for each class. We note that when fine-tuning the models using Layer FT and SCL FT, in particular, we only fine-tune Block 5 of the pre-trained ResNet18 model. We also conduct the same experiment using the transformer-based architecture, for which we fine-tune pre-trained CLIP model using subsets of Cifar-100 dataset with sizes of each class from: [7, 17, 38, 87, 200]. When fine-tuning with Layer FT and SCL FT, we report the results obtained by fine-tuning Layer 8 of the pre-trained CLIP model [14]. We run each experiment reported in the figure 3 times using different random seeds and report the average results.

**Experimental details for Figure 12.** In Figure 12, we pre-train ResNet50 models with different number of projection layers and different loss functions (CE, MSE, SupCon) using Cifar-100 for 200 epochs. Then we use the learned model to do transfer learning on Cifar-10. The $\mathcal{NC}_1$ is then evaluated on the source dataset. For projection head used for different losses, we use MLP with ReLU activation, and we vary the number of its layers.

**Experimental details for Figure 10.** With the same pre-training setup as in Figure 1, we transfer the pre-trained models to four different downstream datasets: Cifar-10, FGVC-Aircraft, DTD, and Oxford-IIIT-Pet. We investigate the relationship between data augmentation/adversarial training and transferability and show the relationship is not universal for the source dataset and transferability.

### E.2 Technical Detail for the Tables

**Experimental details for Table 2.** The negative correlation between downstream $\mathcal{NC}_1$ and transfer accuracy extends to few-shot learning, as shown in Table 2. We observed that the $\mathcal{NC}_1$ of the penultimate layer negatively correlates with the few-shot classification accuracies on miniImageNet and CIFAR-FS meta-test splits. To obtain few-shot accuracy, we followed (Tian et al., 2020) and first learned a feature extractor through supervised learning on the meta-training set. We then trained a linear classifier on top of the representations obtained by the extractor. We calculated the $\mathcal{NC}_1$ of the penultimate layer in the feature extractor network using different architectures and observed that they are negatively correlated with the 1-shot and 5-shot accuracies.

Following previous works (Mishra et al., 2017; Oreshkin et al., 2018), ResNet12 is adopted as our backbone: the network consists of 4 residual blocks, where each has three convolutional layers with $3 \times 3$ kernel; a $2 \times 2$ max-pooling layer is applied after each of the first three blocks; a global average-pooling layer is on top of the fourth block to generate the feature embedding.

On the optimization side, we use SGD optimizer with a momentum of 0.9 and a weight decay of $5e-4$. Each batch consists of 64 samples. The learning rate is initialized as 0.05 and decayed with a factor of 0.1 by three times for all datasets, except for miniImageNet where we only decay twice as the third decay has no effect. We train 100 epochs for miniImageNet, and 90 epochs for both CIFAR-FS. When training the embedding network on transformed meta-training set, we adopt augmentations such as random crop, color jittering, and random horizontal flip as in (Lee et al., 2019). For meta-testing time, we train a softmax regression base classifier. We use the multi-class logistic regression implemented in scikitlearn (Pedregosa et al., 2011) for the base classifier.

---

[13]This method significantly reduces the computational complexity, while it may also introduce a certain degree of imprecision in the $\mathcal{NC}_1$ calculation.

[14]For the CLIP model, We sweep 2 learning rates [1e-2 1e-3] and 2 weight decays [1e-4 0.0] to select the optimal setting for each method on the full Cifar-100 dataset and use the found hyperparameters for all expriments.

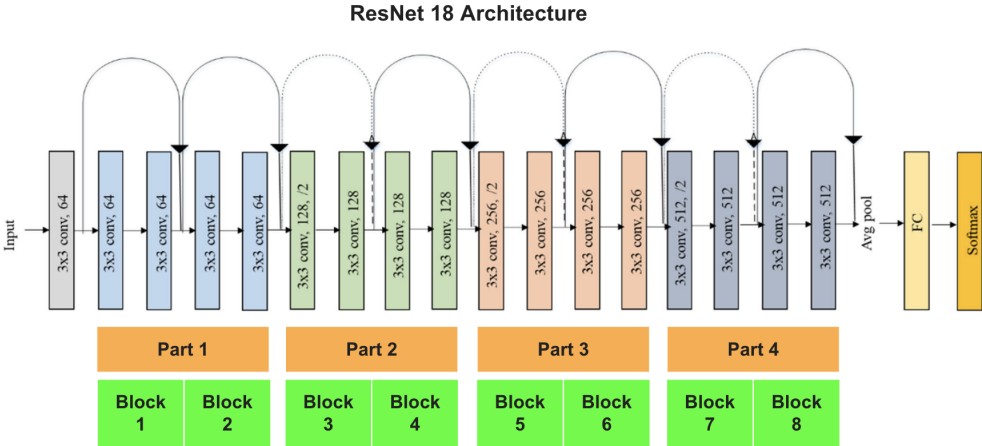

Figure 13: **Fine-tuning unit of ResNet.** Image of ResNet18 from (Ramzan et al., 2019).

**Experimental details for Table 3.** In Table 3, we compare the performance between linear probing, MLP probing, Layer FT, SCL FT and full model FT based upon a wide variety of experimental setups, such as different model architectures, different pre-training and downstream datasets. In particular, we consider the following.

- **Fine-tuning blocks for different network architectures.** Each residual block in the ResNet models is regarded as a fine-tuning unit, and only the first residual block of each channel dimension is fine-tuned. As illustrated in Figure 13, the feature extractor of ResNet18 is partitioned into four parts, with each part containing two residual blocks of the same channel dimension, and the first block of each part is fine-tuned. In the case of the Vit-B32 model, each of its 12 encoder layers is considered a fine-tuning unit.

- **SCF FT for different models.** For SCL FT with skip connections, the Vit-B32 (same architecture is used in CLIP) model had a constant feature dimension across layers, making it easy to directly apply the skip connection between the fine-tuned layer and the penultimate layer features. However, ResNet models have varying numbers of channels and feature dimensions across layers, making it necessary to calculate the skip connection differently. To address this for ResNet, we first perform adaptive average pooling on the fine-tuned layer features to ensure that each channel had only one entry, and then we use zero-padding to match the number of channels with the penultimate layer features. Finally, for both ResNet and ViT models, we apply the skip connection and used the combined features to train the classifier.

- **Batch / Layer normalization.** For a fair comparison between different fine-tuning methods, we always let the normalization layers (Ioffe & Szegedy, 2015; Ba et al., 2016) update the running means and variances on the downstream data for all fine-tuning methods (i.e., linear probing, layer FT, SCL FT, and full model FT).

In terms of hyperparameter selection, we consider 2 learning rates ($[1e-1, 5e-2]$ for ResNet models and $[1e-2, 1e-3]$ for ViT) and 2 weight decays ($[1e-4, 5e-4]$ for ResNet models and $[1e-4, 0.0]$ for ViT) to identify the optimal settings for each method. Due to limited computing resources, we conduct hyperparameter search using the Cifar-10 dataset and use the found optimal setting for all datasets. We report the final selected hyperparameters in Table 4. We run each experiment reported in the table 3 times using different random seeds and report the mean and standard deviation.

**Experimental details for Appendix C.** In Appendix C, we pre-trained ResNet18 models on the Cifar-100 dataset using three different loss functions: CE, MSE, and SupCon, each for 200 epochs. For transfer to the Cifar-10 dataset, we froze the pre-trained models and trained only a linear classifier on top of them for 200 epochs. Both procedures employed the cosine learning rate scheduler with an initial learning rate of 0.1.

Table 4: **Hyperparameter setup for Table 3.**

| | ResNet | | | | | ViT | | | | |
|---|---|---|---|---|---|---|---|---|---|---|
| | Linear probe | MLP probe | Layer FT | SCL FT | Full FT | Linear probe | MLP probe | Layer FT | SCL FT | Full FT |
| Learning rate | $5e-2$ | $5e-2$ | $5e-2$ | $5e-2$ | $5e-2$ | $1e-2$ | $1e-2$ | $1e-2$ | $1e-2$ | $1e-3$ |
| weight decay | $1e-4$ | $5e-4$ | $1e-4$ | $1e-4$ | $5e-4$ | $0$ | $0$ | $0$ | $0$ | $0$ |

Table 5: **Comparison between Layer FT and Layer LoRA on the VTAB-1k benchmark.** We report the average results from 3 different runs.

| | Cifar100 | Caltech101 | DTD | Flowers102 | Pets | SVHN | Sun397 | EuroSAT | Resisc45 | Retinopathy | Clevr-Count | Clevr-Dist | DMLab | dSpr-Loc | dSpr-Ori | sNORB-Azim | sNORB-Ele | Average |
|---|---|---|---|---|---|---|---|---|---|---|---|---|---|---|---|---|---|---|
| Layer FT | 56.7 | 95.0 | 67.3 | 99.7 | 93.5 | 75.3 | 46.8 | 96.8 | 84.7 | 76.7 | 57.8 | 56.7 | 46.3 | 53.3 | 51.8 | 18.2 | 33.8 | **65.3** |
| Layer LoRA | 64.3 | 94.7 | 65.3 | 100.0 | 94.7 | 61.0 | 53.8 | 96.3 | 79.3 | 75.3 | 61.3 | 60.5 | 45.8 | 45.0 | 31.7 | 16.2 | 33.3 | **63.4** |

We computed $\mathcal{NC}_1$ on the source dataset (Cifar-100) and evaluated the downstream dataset's (Cifar-10) test accuracy.

# F    Additional Experimental Results and Discussions

In this final part of the appendices, we present extra complementary experimental results to corroborate our claims in the main body of the paper.

## F.1    Combination of LoRA and Layer FT

LoRA (Hu et al., 2021) is a popular parameter-efficient fine-tuning strategy broadly used in different domains of deep learning. Its core strategy involves integrating an additional low-rank sub-module into all layers of the network during the fine-tuning phase. On the other hand, our method choose a key layer to fine-tune. As such, the two methods serve complementary purposes and can be combined. Therefore, we combine the two approaches by applying LoRA on a single layer (with rank= 96) while freezing the rest, we term this combined approach Layer LoRA. We then use Layer FT and Layer LoRA on the same layer and compare their performances using the VTAB-1k benchmark (Zhai et al., 2019). The results are reported in Table 5. We can observe that although Layer LoRA has a much lower parameter budget, its performance is not significantly lower than Layer FT in this limited data regime. This outcome hints at potential improvement of Layer FT for better parameter efficiency, which we leave as a future direction.

## F.2    Layer selection on ViT models

For ResNet-based models, our experiments demonstrate that fine-tuning layers closer to the classifier consistently yields smaller penultimate $\mathcal{NC}_1$ values and superior transfer performance after fine-tuning. However, our exploration of ViT models reveals a distinct pattern. Specifically, we conduct Layer-FT for various layers and evaluate downstream $\mathcal{NC}$ and transfer accuracy after fine-tuning. According to Figure 14, we can make the following observations: (I) the negative correlation between $\mathcal{NC}_1$ and transfer accuracy persists in these experiments.; (ii) marginal disparity in transfer accuracy performance across layers, except for the initial and final layers; (iii) the most collapsed models are typically obtained by fine-tuning middle-to-late layers. Motivated by these observations and to maintain consistency with our experimental methodology applied to ResNet models, we opt to fine-tune Layer 8 of ViT models throughout our analysis. Notably, this selection consistently yields near-optimal transfer performances in most scenarios.

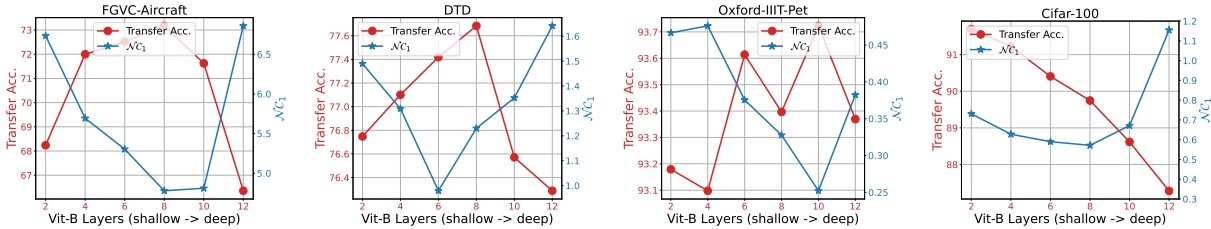

Figure 14: **Transfer accuracy and penultimate $\mathcal{NC}_1$ for ViT models when fine-tuning different layers.** We conduct Layer-FT on the ViT model for different layers and show the downstream transfer accuracy and penultimate $\mathcal{NC}_1$ after fine-tuning.

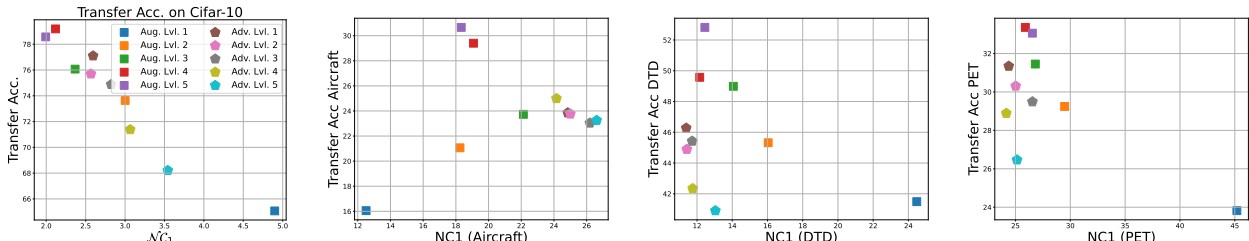

Figure 15: **Transfer accuracy on different downstream tasks and $\mathcal{NC}_1$.** Transfer accuracy and $\mathcal{NC}_1$ are measured on the same downstream datasets.

### F.3 Experiments for Section 3 & Section 4

**Extra experiments for Section 3.** In the main paper's Figure 1, we illustrate a correlation between $\mathcal{NC}_1$ on Cifar-10 and transfer accuracy on various downstream tasks for ResNet50 models pre-trained on Cifar100. However, when evaluating this relationship directly on downstream datasets for ResNet50 models pre-trained on Cifar100, the negative correlation becomes less apparent, as depicted in Figure 15. Nonetheless, we believe that $\mathcal{NC}_1$ is somewhat less sensitive when the magnitude is too large, as shown in Figure 2; when $\mathcal{NC}_1$ has a smaller magnitude, the negative correlation on downstream data becomes clear again.

**Extra experiments for Section 4.** In Figure 7, we only visualized the layer-wise $\mathcal{NC}_1$ for ResNet18 models on downstream dataset FGVC-Aircraft with different fine-tuning methods. Here, we include the same experiments but for DTD and Oxford-IIIT-Pet in Figure 16. Additionally, for a more comprehensive study, we conducted similar experiments for the pre-trained ViT-B32 models, where we use the classification token of each layer to calculate the associate $\mathcal{NC}_1$. In Figure 17, we observe a similar trend that we have seen on ResNet18 models, that within-class variability of the ViT model (measured by $\mathcal{NC}_1$) decreases from shallow to deep layers and the minimum values often appear near the penultimate layers. Moreover, we can observe that Layer FT and SCL FT always have smaller penultimate $\mathcal{NC}_1$ compared with Linear Probing and also better transfer performance.

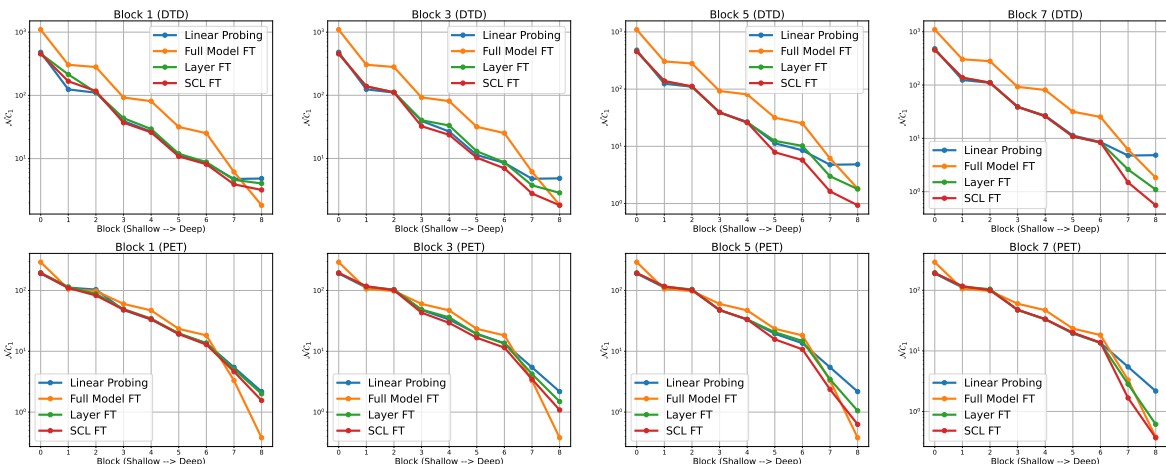

Figure 16: **Layer-wise $\mathcal{NC}_1$ for different fine-tuning methods on ImageNet-1k pre-trained ResNet18.** We compare the layer-wise $\mathcal{NC}_1$ across different fine-tuning methods, including linear probing, Layer FT, and SCL FT, using the ImageNet-1k pre-trained ResNet18 backbone. We evaluate the models on the downstream datasets DTD (Top), and Oxford-IIIT-Pet (bottom). For the figures from left to right, we plot the results of fine-tuning only Block 1, Block 3, Block 5, and Block 7 in both Layer FT and SCL FT.

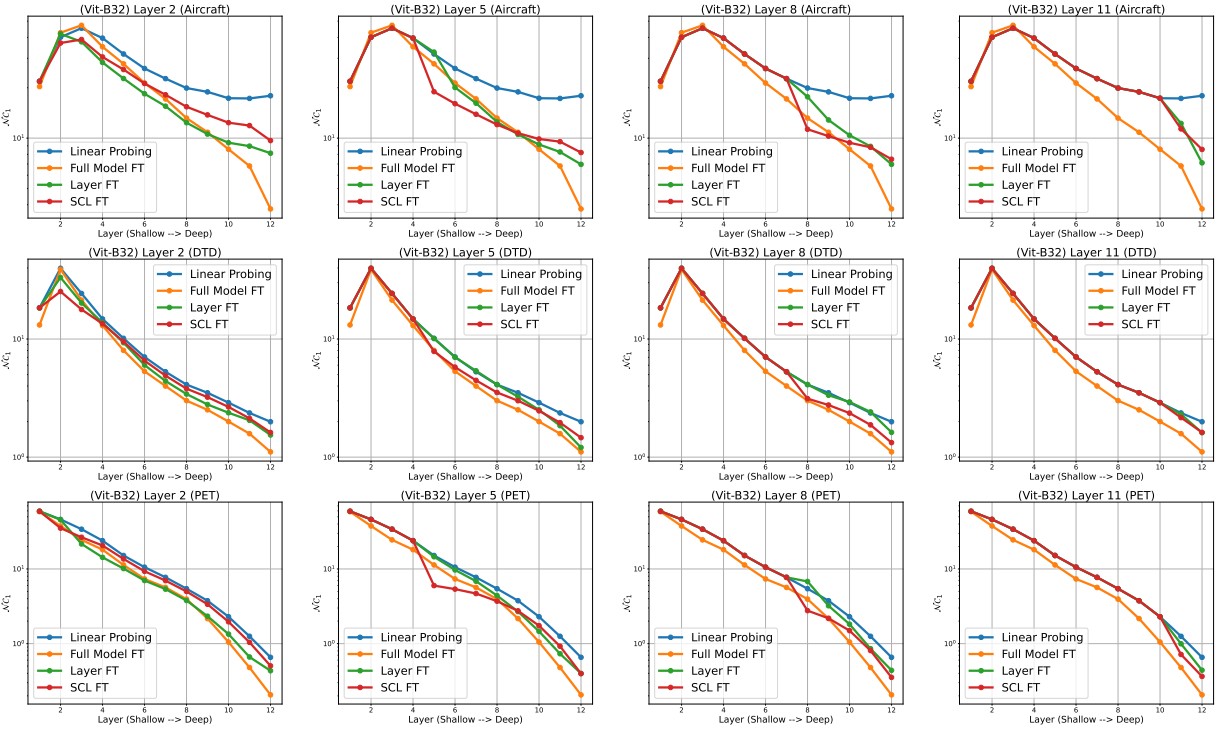

Figure 17: **Layer-wise $\mathcal{NC}_1$ for different fine-tuning methods.** We compare the layer-wise $\mathcal{NC}_1$ across different fine-tuning methods, including linear probing, Layer FT, and SCL FT, using the pre-trained Vit-B32 model. We evaluate the models on the downstream datasets of FGVC-Aircraft (top), DTD (middle), and Oxford-IIIT-Pet (bottom). For the figures from left to right, we plot the results of fine-tuning only Layer 2, Layer 5, Layer 8, and Layer 11 in both Layer FT and SCL FT.

