# OpenReview forum: "Understanding and Improving Transfer Learning of Deep Models via Neural Collapse"
_TMLR — Accepted by TMLR_

### Review · Reviewer_aafv · 2024-02-12

**Summary Of Contributions:**

This paper studies the connections between transfer learning performance an degree of neural collapse of different layers of a neural network. Specifically, it argues that more collapsed features, from the point of view of the fine-tuning data, lead to better fine-tuned performance. The paper then presents a new fine-tuning technique consisting on running linear probing over a linear mixture of the final layer features and a hand-picked layer deeper in the network (chosen based on the previous neural collapse heuristic). The results show sometimes a marginal improvement over just linear probing over the deeper hand-picked layer. Finally, the paper discusses the results of some ablation studies fine-tuning CIFAR-10 starting from models pre-trained on CIFAR-100 with different settings.

**Audience:**

Yes

**Broader Impact Concerns:**

I see no broader impact concerns stemming from this work.

**Claims And Evidence:**

No

**Requested Changes:**

The paper requires substantial changes in my opinion in order to be accepted, probably beyond what is possible on a minor revision to the manuscript. In particular, in order to support the generality claims of the authors, I would need to see a significant revamp of the experimental results including repeating most experiments over a consistent suite of diverse datasets, pre-trained checkpoints, and training conditions. The experiments would need to be better structured and replicated to isolate the effect of different factors in the results. Furthermore, all the unclear points that I mentioned in the Weaknesses should be addressed properly, as right now, these unclear points make the whole premise of the paper inconsistent.

**Strengths And Weaknesses:**

## Strengths

1. **Promising heuristic**: I find using a neural collapse heuristic to select the best layer to fine-tune on a model and interesting idea worth exploring.
2. **Interesting observations**: The trends of Figure 2, 3, and 4 seem consistent and interesting.

## Weaknesses

1. **Lack of thorough experimentation**: Overall, I believe that most claims in Sections 4 and 5 lack proper empirical validation. For example, the paper makes significant statements regarding the improvements of their new method that are not substantiated by the marginal and inconsistent improvements in Table 2. Moreover, the ablation studies of Section 5 have only been performed on a nowadays toyish setting (fine-tuning a network on CIFAR-10 after pre-training on CIFAR-100) without proper repetitions. Still the observed trends are  extrapolated in writing to arrive at ver generic conclusions by the authors without proper evidence. I highlight that I am not against performing experiments on simple an low-compute settings. My main concern is with claiming that those experiments can say anything about more realistic settings (like fine-tuning a CLIP model on some video retrieval task) without proper evidence.
2. **Unclear how neural collapse is computed**: The paper is unclear in which features are used in Eq. (4) to estimate neural collapse. Are thy the pre-trained features of the model over the fine-tuned data, or the fine-tuned features over the fine-tuned data? If the former, how is it possible that different fine-tuning techniques yield different levels of neural collapse? And if the latter, how is it possible that methods yielding the lowest level of neural collapse in Table 2 do not lead to the best performances?
3. **Hyperparameter tuning strategy is unclear**: The paper does not mention what strategy was used to tune the different experiments. In this regard, I am worried that most results and comparisons with other baselines may be obscured by a lack of proper tuning.

---

> ### Author Response · Authors · 2024-03-20
> **Response to Reviewer aafv (1/2)**
>
> We thank the reviewer for the careful review of our paper and questions. In the following, we address the reviewer’s comments one by one.
>
> >The ablation studies of Section 5 have only been performed on a nowadays toyish setting (fine-tuning a network on CIFAR-10 after pre-training on CIFAR-100) without proper repetitions. Still the observed trends are extrapolated in writing to arrive at ver generic conclusions by the authors without proper evidence. I highlight that I am not against performing experiments on simple an low-compute settings. My main concern is with claiming that those experiments can say anything about more realistic settings (like fine-tuning a CLIP model on some video retrieval task) without proper evidence.
>
> We thank the reviewer for the suggestions to improve our paper. First, we want to make the following clarifications:
>
> * We want to emphasize that the main focus of the study is on the relationship between NC and transfer learning accuracy on the downstream tasks, where in Section 3 & 4 our study is comprehensive covering the results of pre-trained models on ImageNet and ViT models.
>
> * Given that there are many existing works on studying the relationship between transfer accuracy and feature diversity on source training data [1,2,3,4], our study in Section 5 is not our major focus. Nonetheless, it shows the comprehensiveness of our study, and it provides complimentary viewpoints on the limitation of using feature diversity on source data in predicting transfer accuracy. Although the experiment is done on a small Cifar dataset, the existence of such an example is sufficient to support the claim.
> In the revision, according to the reviewer’s concerns, we only maintain our observation on the limitations of source NC in terms of measuring transfer accuracy in the main body of the paper and move the rest of Section 5 to Appendix A. We also include a limitation discussion regarding our small experimental scale in the corresponding text.
>
> We hope this can address the reviewer’s concerns, and willing to further modify our claims and add more experiments if needed, etc.
>
> [1] Kornblith et al; Why do better loss functions lead to less transferable features?
>
> [2] Galanti et al; On the Role of Neural Collapse in Transfer Learning
>
> [3] Feng et al; Rethinking Supervised Pre-training for Better Downstream Transferring
>
> [4] Nayman et al; Diverse Imagenet Models Transfer Better
>
> > Unclear how neural collapse is computed: The paper is unclear in which features are used in Eq. (4) to estimate neural collapse. Are they the pre-trained features of the model over the fine-tuned data, or the fine-tuned features over the fine-tuned data? If the former, how is it possible that different fine-tuning techniques yield different levels of neural collapse?
>
> We thank the reviewer for raising this question. We make the following clarification:
>
> * For Section 3, we estimate NC on the downstream data with pre-trained model frozen. We evaluate NC on the last-layer feature as well as intermediate layers
> * For Section 4, we fine-tune pre-trained model and evaluate NC on the last layer of the fine-tuned model on the downstream data. Hence different fine-tuning methods induce different levels of NC.
> * For Section 5, we evaluate NC on the source training data when the model is trained until convergence.
>
> To make the above points clear, we created a table for summary on Page 5.
>
> > And if the latter, how is it possible that methods yielding the lowest level of neural collapse in Table 2 do not lead to the best performances?
>
> The new results in Table 3 suggest that lower NC in the last layer typically leads to better transfer accuracy with a few exceptions. For these outliers, we hypothesize that potential overfitting could be compromising the transfer accuracy.

---

> ### Author Response · Authors · 2024-03-20
> **Response to reviewer aafv (2/2)**
>
> > Lack of thorough experimentation: Overall, I believe that most claims in Sections 4 and 5 lack proper empirical validation. For example, the paper makes significant statements regarding the improvements of their new method that are not substantiated by the marginal and inconsistent improvements in Table 2. The paper does not mention what strategy was used to tune the different experiments. In this regard, I am worried that most results and comparisons with other baselines may be obscured by a lack of proper tuning.
>
> We thank the reviewer for raising this question. Per the reviewer’s suggestions, in Section 4, we reconducted our experiments in the following ways.
>
> * **Fine-tuning learning rates and weight decay.** we carefully tuned the learning rates ([1e-1 5e-2] for ResNet and [1e-2 1e-3] for ViT / CLIP) and weight decays ([1e-4 5e-4] for ResNet and [1e-4 0.0] for ViT / CLIP) to identify the optimal parameters for each method.
>
> * **Error bar over multiple random seeds.** For each case, we run our experiments over 3 random seeds, and show the average as well as the variance.
>
> The updated results are shown in Table 3 and Figure 7 in Section 4, and the experimental setup is now discussed in Appendix D. We observe from Table 3 that all methods benefit from the hyperparameter tuning, but the general trend remains similar with the previous version that our proposed methods outperforms linear probing by a great margin and achieves on-par performance with full model FT in some scenarios. Furthermore, the results in Figure 7 indicate the robustness of our proposed methods in terms of data scarcity. We refer the reviewer to the updated Section 4 and Appendix D for more details.

---

### Review · Reviewer_D8Kd · 2024-02-17

**Summary Of Contributions:**

This paper studies the relationship of "neural collapse" (NC) with transfer learning accuracy on downstream tasks, comparing linear classifiers on different features, end-to-end finetuned models and an introduced technique, "skip-connection layer" tuning.  More collapse is correlated with better accuracy, with both measured on the target task.  However, when measured on the source data, the opposite is observed, with higher accuracy is reached at relatively low collapse, and there is a peak where best accuracy is observed.  Skip-connection tuning is an interesting construction, training only one intermediate layer with both direct and layered connections to the classifier, and achieves good performance on a variety of task metrics.

**Audience:**

No

**Claims And Evidence:**

Yes

**Requested Changes:**

All critical:
* Explanation of what, if any, insights are provided by measuring NC on the target data.  Or alternatively, deemphasizing these sections.
* Comparisons between SCL FT and a wider selection of simple transfer learning constructions.  In particular, comparing to unfreezing more layers, and to small MLP (rather than linear) on top of intermediate layers (penultimate or same layer as chosen for SCL).
* Clearer explanation of the relation between NC, and SCL FT motivations.  I don't understand how the observations in sec 3 for NC are linked to inspiration for the skip-connection, as described on p.8.  Is there a reason the skip connection in particular (as opposed to other alternatives) is motivated by NC measurements?

Less critical but would strengthen work:
* Additional investigation of the peak value NC on source data --- why this particular value?

**Strengths And Weaknesses:**

Unless I'm missing something, the study of "neural collapse" applied to the target task is uneventful.  The definition of NC measures a combination of covariances to incorporate within-class similarity and between-class difference.  In particular, as classes become more separate, NC is higher.  So the main thrust seems to boil down to comparing one measure of class separability (NC) with another (linear classification), and saying they relate.  This is not a very useful conclusion.

Things get more interesting in the other two pieces of this paper, skip-connection tuning and the study of NC on feature diversity.  However, both appear underdeveloped.  SCL FT is an interesting construction I don't think I've seen in this exact form before, but it isn't compared with more vanilla constructions like varying the number of layers being trained, or a linear classifier or 2- or 3-layer MLP/convnet starting from different feature layers.  SCL FT is basically using both an MLP (original path) and linear classifier (skip-conn path), with MLP being frozen to the original model.  What are the particular advantages of this (if any) compared to smaller MLP or unfreezing some more weights?  In addition, while this paper cites Kumar &al 2022, I didn't see a comparison to their 2-stage method of linear followed by full fine-tuning, which would be another good point of comparison.

The study of NC measured on the source data starts to show some interesting behaviors, but doesn't go very far in exploring them.  The fact that there there is a peak value for target task accuracy vs source NC (and near 1.0, no less) is curious, but overall expected:  in the limiting cases, maximum collapse happens when every datapoint maps to one of K points (for K source classes), so target classes would be distinguishable only to the point where they correlate with single source classes.  At the other extreme, as the authors also describe, completely random features would have low collapse.  So best target classifier accuracy ought to happen at an intermediate value, as was found.  But the fact that this value is right at 1 is a little suggestive --- it might be a coincidence, but to me it seems it may have to do with a balanced data distribution and structure of downstream classifier.  Is there a reason it would peak at this particular value?  What about for other data distributions or classifier heads?  It's possible this route might go too far astray, but to me some additional investigation here looks it could be worthwhile.

---

> ### Comment · Reviewer_D8Kd · 2024-03-08
> **clarification**
>
> After looking the reviews here, I realized my comments on the target data NC measurements may have been a little too short and without enough explanation on why I didn't find the measurements practical, so wanted to explain a bit more so that my questions about that are clearer.
>
> The reason I didn't find the application on NC on target data very useful, was NC appears to measure class separation in a way that indicates how well the classifier could perform, but still requires the target data and labels, and doesn't seem necessarily that much cheaper to compute compared to training the classifier itself.  Indeed, there are comments in the text around the matrix inverse being expensive, so that there are approximations.  So if computing NC requires all the same data and labels as training the classifier, and it isn't very cheap to compute, why not just train the classifier itself instead and measure its performance directly, then choose the best layer based on that?  What is the computational cost of NC vs classifier training, or are there any other advantages I missed?

---

> ### Author Response · Authors · 2024-03-20
> **Response to Reviewer D8Kd (1/3)**
>
> We thank the reviewer for the careful review of our paper and questions. In the following, we address the reviewer’s comments one by one.
>
> > Unless I'm missing something, the study of "neural collapse" applied to the target task is uneventful. The definition of NC measures a combination of covariances to incorporate within-class similarity and between-class difference. In particular, as classes become more separate, NC is higher. So the main thrust seems to boil down to comparing one measure of class separability (NC) with another (linear classification), and saying they relate. This is not a very useful conclusion. The reason I didn't find the application on NC on target data very useful, was NC appears to measure class separation in a way that indicates how well the classifier could perform, but still requires the target data and labels, and doesn't seem necessarily that much cheaper to compute compared to training the classifier itself. Indeed, there are comments in the text around the matrix inverse being expensive, so that there are approximations. So if computing NC requires all the same data and labels as training the classifier, and it isn't very cheap to compute, why not just train the classifier itself instead and measure its performance directly, then choose the best layer based on that? What is the computational cost of NC vs classifier training, or are there any other advantages I missed?
>
> We thank the reviewer for the critical question, and we want to make some clarification on NC1 and the understanding of our work.
>
> * **The differences between NC1 and class separability.** First, as discussed in [1] and [2], the metric NC1 is mainly to quantify the collapse of within-class variability (or in other words, feature diversity) relative to the between-class separation. As such, it is important to distinguish that training a linear classifier operates on a fundamentally different principle than NC1. While the former aims to discriminate between classes, the latter mainly assesses the extent of variability collapse within each class.
>
> * **Linear classification accuracy on downstream training data might not be a good indication of downstream test performance.** From classical machine learning theory, we know that good training accuracy on the downstream data does not have good generalization performance. We might overfit on limited downstream training data but not generalize to the downstream test data. One must test the trained linear classifier on a held-out test set to know a model’s actual performance. In contrast, NC1 can provide valuable insights into a model's transfer performance based solely on the training features from a downstream task. To further demonstrate this point, we conducted an experiment for which we used the image encoder of the CLIP model as a feature extractor to extract training and testing features from multiple downstream datasets. We train linear classifiers evaluate NC on training features and then calculate training/testing accuracy using the fitted linear classifier. The plots of NC1 vs transfer accuracy and training accuracy vs. transfer accuracy are shown in Figure 11 (Appendix E.1). We observe that the negative correlation between NC and transfer accuracy persists, while no obvious correlation can be observed for the training accuracy. We refer the reviewer to Appendix E.1 for more detail.
>
> * **Our fine-tuning method in Section 4 does not need to evaluate NC1 for every layer.** First, although we agree with the reviewer that the evaluation of NC1 could become expensive if the feature dimension is too large, our investigation in Section 3 was solely aimed at exploring the correlation between the downstream NC1 and transfer accuracy, but not for direct practical usage. The insight gained from Section 3 underpins the intuition for our SCL fine-tuning strategy introduced in Section 4, which aims to minimize the last-layer NC for improved transfer effectiveness. Consequently, in practical applications, it's not necessary to compute NC1 when using our SCL fine-tuning technique.
>
>
> [1] Papyan et al; Prevalence of neural collapse during the terminal phase of deep learning training
>
> [2] Zhu et al; A Geometric Analysis of Neural Collapse with Unconstrained Features

---

> > ### Comment · Reviewer_D8Kd · 2024-04-09
> > **responses**
> >
> > Thanks for your responses and the new revision.  They appear to have addressed many of the questions raised, and in particular, from my review, the following are mostly addressed now:
> >
> > Fig 11 in the Appendix is much more convincing in showing the benefit of NC1 vs direct classifier measurement. While a direct accuracy measurement of the classifier might also be measured on a validation set, and the val set measure could have better correlation to test accuracy, the fact that the NC1 measure does not need a split-out validation set to get this degree of correlation is appealing.
> >
> > Fig 13 is also good, showing the performance and params tuned for a simple unfreezing baseline. It would be helpful to also include the results (test perf and params tuned) for SCL-FT here by comparison.
> >
> > Fig 12 is interesting, but is it consistent with the findings in Fig 6? These two figures use different architectures and different datasets. But Fig 6 shows that adapting the last layers ought to work best, while Fig 12 is the first (layer 2). What is the NC1 metric for the points in Fig 12?
> >
> >
> > At this point, as you also mention in your comments, the paper (now with extended appendix) is somewhat less focused and more scattered.  Incorporating some of the highlights of the newer text into the main text may help here.

---

> > > ### Author Response · Authors · 2024-04-14
> > > **Response to Reviewer D8Kd**
> > >
> > > We are truly grateful to the reviewer for the thoughtful comments and appreciation of our rebuttal. We also appreciate the additional questions provided, which have guided further improvements to our paper. In the revision, we have addressed these new questions and made further updates, detailed in the main body and Appendix E.5.
> > >
> > > > Fig 13 is also good, showing the performance and params tuned for a simple unfreezing baseline. It would be helpful to also include the results (test perf and params tuned) for SCL-FT here by comparison.
> > >
> > > We thank the reviewer for the suggestion. In the revision, we have updated Figure 13 to include the SCL FT results. Since SCL FT does not introduce auxiliary parameters, the parameter count remains the same with Layer FT. Additionally, SCL FT shares the same trend with Layer FT with slightly improved performance.
> > >
> > > > Fig 12 is interesting, but is it consistent with the findings in Fig 6? These two figures use different architectures and different datasets. But Fig 6 shows that adapting the last layers ought to work best, while Fig 12 is the first (layer 2). What is the NC1 metric for the points in Fig 12?
> > >
> > > We thank the reviewer for the careful reading of our paper and the question. As the reviewer noticed, this is due to the fact that the network architectures used in Figure 6 (ResNet) and Figure 12 (ViT) are different.
> > >
> > > In our study, we find that the trend of NC1 across layers for different architectures (i.e., ResNet vs. ViT) are slightly different, resulting in different optimal fine-tuning layers. Specifically, for ResNet, the lowest NC1 consistently emerges in the layer close to the classifier. In contrast, for ViT, the lowest NC1 often emerges in the middle to late layers with larger variability, but the negative correlation between NC1 and transfer accuracy persists. To demonstrate this point, we have included an experiment where we applied Layer FT for different layers of the ViT model and then evaluated the penultimate NC1 along with the transfer accuracy. These findings are presented in Figure 14 (Appendix E.5), where most neural collapsed models are typically obtained by fine-tuning middle-to-late layers.
> > >
> > > Motivated by these observations and to strike a balance between parameter efficiency, we choose to fine-tune a middle-to-late layer for both types of architectures. For ResNet18, we fine-tune the 5-th block, and for ViT we fine-tune Layer 8.  We found this choice is robust across different dataset, and typically yields near-optimal results.
> > >
> > > > At this point, as you also mention in your comments, the paper (now with extended appendix) is somewhat less focused and more scattered. Incorporating some of the highlights of the newer text into the main text may help here.
> > >
> > > We thank the reviewer for the great suggestion. In the revision, we moved the discussion regarding the benefit of NC1 vs direct classifier measurement, which addresses the reviewer’s major concern, to the main body of the paper (we plan to, but haven’t moved Figure 11 to the main body yet since this will change all the figure indices which may cause inconvenience for our discussion with the reviewers).
> > > Per the reviewer’s suggestion, in the final version, we plan to further re-organize the paper based on the reviewers’ suggestions and further move other key results in Appendix E that supporting our claims into the mainbody.

---

> ### Author Response · Authors · 2024-03-20
> **Response to Reviewer D8Kd (2/3)**
>
> > Clearer explanation of the relation between NC, and SCL FT motivations. I don't understand how the observations in sec 3 for NC are linked to inspiration for the skip-connection, as described on p.8. Is there a reason the skip connection in particular (as opposed to other alternatives) is motivated by NC measurements?
>
> Following the previous question, in Section 3, we empirically validate the presence of a general negative correlation between the last-layer NC1 measured on downstream tasks and the corresponding transfer accuracy. This observation leads us to hypothesize that better transfer accuracy can be achieved by reducing the last-layer NC1 on downstream data.
>
> Intuitively, as discussed on the 2nd bullet point of Page 9, the addition of a skip connection enables the model to directly transmit fine-tuned layer feature information to the classifier, bypassing other intermediate layers, the direct link between the classifier and fine-tuned feature could be advantageous in reducing the magnitude of the NC1 metric in the last-layer. Our observation in Figure 6 verifies this intuition, and the results in Table 3 demonstrate the effectiveness of our approach.
>
> > The study of NC measured on the source data starts to show some interesting behaviors, but doesn't go very far in exploring them. The fact that there there is a peak value for target task accuracy vs source NC (and near 1.0, no less) is curious, but overall expected: in the limiting cases, maximum collapse happens when every datapoint maps to one of K points (for K source classes), so target classes would be distinguishable only to the point where they correlate with single source classes. At the other extreme, as the authors also describe, completely random features would have low collapse. So best target classifier accuracy ought to happen at an intermediate value, as was found. But the fact that this value is right at 1 is a little suggestive --- it might be a coincidence, but to me it seems it may have to do with a balanced data distribution and structure of downstream classifier. Is there a reason it would peak at this particular value? What about for other data distributions or classifier heads? It's possible this route might go too far astray, but to me some additional investigation here looks it could be worthwhile.
>
> We thank the reviewer for the insights. The main purpose of Section 5 is to study the relationship between feature collapse (NC1) and the transfer accuracy on the source training data. Given that there are quite a few existing studies in this setting (e.g., different training losses [1,2], different metrics of feature diversity [3,4]), we provide new findings on the role of projection head and data augmentation, and we also give a counter-example to show the limitation of using feature diversity as an indicator for transfer learning performance.
>
> Figure 9 demonstrates that increasing feature diversity does not necessarily improve transfer accuracy. This phenomenon could be attributed to a tradeoff between feature diversity and class separation, which may explain the peak observed in Figure 8, as the reviewer has noted. However, we speculate that the peak value around 1.0 in Figure 8 might be coincidental, especially since Figure 9 (Left) shows a sustained positive correlation even when the pre-train NC surpasses 2.0. Nevertheless, we agree with the reviewer that examining the tradeoff between feature diversity and class separation in the context of transfer learning presents an interesting direction for future research.
>
> In our revision, we have included discussions of this peak point and possible future directions in Page 11 (Section 5).
>
> [1] Kornblith et al; Why do better loss functions lead to less transferable features?
>
> [3] Islam et al; A Broad Study on the Transferability of Visual Representations with Contrastive Learning
>
> [3] Galanti et al; On the Role of Neural Collapse in Transfer Learning
>
> [4] Nayman et al; Diverse Imagenet Models Transfer Better

---

> ### Author Response · Authors · 2024-03-20
> **Response to Reviewer D8Kd (3/3)**
>
> > Comparisons between SCL FT and a wider selection of simple transfer learning constructions. In particular, comparing to unfreezing more layers, and to small MLP (rather than linear) on top of intermediate layers (penultimate or same layer as chosen for SCL).
>
> We thank the reviewer for suggesting additional comparisons.
>
> * In revision, we have implemented MLP probing—replacing the final linear classifier with a trainable 2-layer MLP classifier with a hidden dimension of 1024—and compared its performance with other methods in Table 3. Our findings indicate that while MLP probing offers a modest improvement over linear probing, the performance still significantly lags behind other methods.
>
> * Regarding the suggestion to unfreeze more layers, we have conducted an ablation study where we fine-tune pre-trained models with varying numbers of unfrozen layers. We report the results in Figure 13 (Appendix E.3). Our observations reveal that unfreezing additional layers results in only a marginal enhancement of performance while sacrificing parameter efficiency. Therefore, in pursuit of a balance between parameter efficiency and transfer learning performance, we choose to fine-tune only one layer in our work. We refer the reviewer to Appendix E.3 for more details.

---

### Review · Reviewer_V8Ro · 2024-03-07

**Summary Of Contributions:**

This paper studies the correlation between neural collapse (NC) metrics and the transfer learning performance of pre-trained classification models. The paper suggests that NC metrics measured on the downstream task are correlated with better transfer learning performance, and that this is a better way to ascertain transfer learning accuracy than NC metrics measured on the source task data. The authors also propose an alternative fine-tuning technique to linear probe fine-tuning that is comparable to full model fine-tuning while being more parameter efficient.

**Audience:**

Yes

**Claims And Evidence:**

Yes

**Requested Changes:**

I would like to see responses to the questions I laid out in my "Concerns" section. If they have already been addressed in the text of the paper, please point me to where they are addressed.

**Strengths And Weaknesses:**

Strengths: Experiments are done on a range of downstream tasks and source (pre-training) tasks. The proposed fine-tuning method also seems to satisfy the claims of similar performance to full model finetuning while being parameter efficient. The authors demonstration that NC metrics measured on downstream tasks are a better indicator of transferability helps clear up confusion regarding whether NC is useful for transfer learning

Concerns:
1. It is unclear whether the NC metrics are computed before or after finetuning on the downstream task. If the metrics are computed before finetuning this is a useful finding. If the NC metrics are computed after finetuning then this is a less surprising finding and not very useful.

2. All experiments seem to involve pre-training classifiers on larger image classification tasks. Demonstrating the authors' observation for models that are trained in a self-supervised or unsupervised manner would be much more interesting since this would shed light on how foundation models that are trained without labels function.

3. For the authors' proposed finetuning method, the authors do not compare how efficient their approach is when compared to LoRA finetuning. This would be an important comparison since LoRA is a popular finetuning method that is also parameter efficient.

4. In the authors' proposed finetuning method that involves identifying the right layer to train, it seems as though you need to run experiments to determine which layer is best suited for finetuning. Is this not fairly expensive to determine?

5. If smaller NC1 is better correlated with transfer accuracy, and progressive collapse indicates that the last layers will have more collapse than previous layers, does this not indicate that the best layer to finetune is the last layer?

6. If the best layer to finetune is below the last layer, does one really need to retain the subsequent layers in the network? It would seem that those layers are redundant?

7. Have you tried finetuning through a simplex ETF layer rather than through a skip connection? This approach would integrate more NC insights into finetuning.

---

> ### Author Response · Authors · 2024-03-20
> **Response to Reviewer V8Ro (1/2)**
>
> We thank the reviewer for the careful review of our paper and questions. In the following, we address the reviewer’s comments one by one.
>
> > It is unclear whether the NC metrics are computed before or after finetuning on the downstream task. If the metrics are computed before finetuning this is a useful finding. If the NC metrics are computed after finetuning then this is a less surprising finding and not very useful.
>
> We thank the reviewer for bringing up this question. In our revision, we made up a new table (i.e. Table 1) in Section 2 to clarify how NC is evaluated to address the reviewer’s concerns.
>
> Specifically, we evaluate NC on frozen pre-trained models on downstream data in Section 3.
> We evaluate NC on fine-tuned models on downstream data in Section 4. In Section 5, we pretrain the models on source data, and evaluate NC on the source data for these models trained until the terminal phase.
>
> > All experiments seem to involve pre-training classifiers on larger image classification tasks. Demonstrating the authors' observation for models that are trained in a self-supervised or unsupervised manner would be much more interesting since this would shed light on how foundation models that are trained without labels function.
>
> We thank the reviewer for raising this very interesting point. In the revision, we conducted extra experiments on CLIP foundation models, which are trained unsupervised by using images and matching captions without label information. In Figure 11(a), we computed NC1 on different downstream datasets along with testing transfer accuracy by only linear probing the pre-trained CLIP model. The negative correlation between downstream NC and transfer accuracy shown in the plot aligns with our observations in Section 3. We refer the reviewer to Appendix E.1 for more detail.
>
> > If smaller NC1 is better correlated with transfer accuracy, and progressive collapse indicates that the last layers will have more collapse than previous layers, does this not indicate that the best layer to finetune is the last layer?
>
> We thank the reviewer for bringing up this question. We want to clarify that the phenomenon of progressive neural collapse, as documented in the literature ([1], [2]), is observed on the source dataset where the model is pre-trained until the terminal phase. For well-trained models, it is noted that the metric NC1 measured on the source dataset tends to decrease progressively with the depth of the layers. However, due to distribution shift and domain gap, this is not true when we measure NC1 of pre-trained models on downstream data without fine-tuning. This divergence is illustrated in Figure 6, Figure 15, and Figure 16 (as indicated by the linear probing curve).
>
> In the revision, we have made this clear in the related work section (See the updated text on Page 3).
>
> [1] He et al; A Law of Data Separation in Deep Learning
>
> [2] Peng et al; Understanding Deep Representation Learning via Layerwise Feature Compression and Discrimination
>
> > In the authors' proposed finetuning method that involves identifying the right layer to train, it seems as though you need to run experiments to determine which layer is best suited for finetuning. Is this not fairly expensive to determine?
>
> We express our gratitude to the reviewer for bringing up this question. Following the question above, in assessing Neural Collapse (NC) across different layers of pre-trained models using the source dataset, it's observed that NC tends to collapse progressively from the initial layers to the deeper ones. Nonetheless, this pattern does not hold when evaluating NC of pre-trained models on downstream data, due to shifts in distribution and the domain gap. This divergence is illustrated in Figure 6, Figure 15, and Figure 16 (as indicated by the linear probing curve). Specifically, with downstream data, the lowest level of NC is often found in one of the layers ranging from middle to deep, beyond which NC starts to increase again, continuing up to the final layer.
>
> As such, our intuition is to choose one of the layers close to the one with the lowest NC, which typically occurs in middle to deep layers. Hence, as written in the last paragraph of Page 8, for ResNet18, we choose to fine-tune Block 5 and for ResNet50, we choose Block 8;  regarding ViT, we select Layer 8 (out of 12 layers) for fine-tuning in all ViT experiments. In practice, we choose these specified layers for evaluating across all downstream tasks, and the choice demonstrates robustness across all evaluations.

---

> ### Author Response · Authors · 2024-03-20
> **Response to Reviewer V8Ro (2/2)**
>
> > If the best layer to finetune is below the last layer, does one really need to retain the subsequent layers in the network? It would seem that those layers are redundant?
>
> We thank the reviewer for the interesting point, we have conducted experiments to show that subsequent layers are essential to achieve good performance in the Layer FT regime. Specifically, we utilized the ViT model on the CIFAR-100 dataset, adopting a setup where only a chosen layer and the final classifier are trained, with all intermediate layers deleted. The outcomes of this experimental setup, which we termed as Layer LP, alongside those of Layer FT, are depicted in Figure 12 (Appendix E.2). The results clearly demonstrate that removing these subsequent layers leads to a significant drop in performance, irrespective of the layer selected for fine-tuning. This suggests that the subsequent layers are essential for extracting information for classification. We refer the reviewer to Appendix E.2 for more details.
>
> > For the authors' proposed finetuning method, the authors do not compare how efficient their approach is when compared to LoRA finetuning. This would be an important comparison since LoRA is a popular finetuning method that is also parameter efficient.
>
> We thank the reviewer for raising the point. Our method tries to find the best layer for fine-tuning, while LoRA adds low-rank sub-modules to all layers for fine-tuning. As such, the two methods serve complementary purposes and can be combined. In our revision, we have added an experiment to incorporate LoRA into our proposed method. To be more specific, we apply LoRA to a single layer on ViT, we refer to this modified LoRA method as Layer LoRA. We also conduct Layer FT on the same layer and we compare the performances of Layer LoRA with Layer FT using the Vtab-1k benchmark, the detailed result is reported in Table 6 (Appendix E.4). We can observe that although Layer LoRA operates on a much lower parameter budget, its performance is not significantly lower than Layer FT in this limited data regime. Such observations inspire us that our methods can be further improved for better parameter efficiency. We leave this for future work and thank the reviewer for the great suggestion.
>
> > Have you tried finetuning through a simplex ETF layer rather than through a skip connection? This approach would integrate more NC insights into fine-tuning.
>
> We thank the reviewer for the question. However, it is unclear to us what the simplex ETF layer is.  Does this imply that using a frozen ETF classifier and fine-tune one of the selected layers via Layer FT or Layer SCL FT? Or does this mean the addition of a frozen ETF classifier on top of the fine-tuned layer, with the intermediate layers being removed?
>
> We are happy to test out and revise our paper according upon the clarification.

---

### Author Response · Authors · 2024-03-20
**Response to all reviewers**

We express our sincere gratitude to all reviewers for their insightful comments and suggestions, which have helped us greatly to improve the quality of our paper. In summary, our paper focuses on investigating the relationship between Neural Collapse (NC) and transfer learning, for which we found that there exists a strong correlation between NC evaluated on the downstream dataset (before fine-tuning) and the corresponding accuracy. The relationship further motivates us to propose parameter-efficient fine-tuning methods for transfer learning.

We appreciate that the reviewers found that (i) our experiments are comprehensive and the study of NC is valuable for understanding transfer learning (Reviewer V8Ro), (ii) the idea of skip-connection fine-tuning is interesting (Reviewer D8Kd), and (iii) the relationship between NC and transfer accuracy on downstream tasks is consistent and interesting (Reviewer aafv).

In the rebuttal, we tried our best to address all the concerns of each reviewer and clarify the understanding of our paper. In response to the comments and suggestions raised by the reviewers, we have carefully revised and added additional results to our paper. All revisions in our paper are marked in blue color. We note that for now, all additional experiments recommended by reviewers are located in Appendix E for better coherence and clarity. After the rebuttal period, we’ll update the main body of the paper and Appendix E according to reviews’ suggestions and better placement of these experiments.

---

### Author Response · Authors · 2024-04-23

We express our sincere gratitude to all the reviewer again for generously dedicating their time and effort to review our paper. The insightful comments and feedback provided have been invaluable in refining our work. As we remain committed to further improve the quality of our paper, we welcome any additional questions or suggestions the reviewers may have to offer. Your continued engagement is deeply appreciated, and we eagerly anticipate hearing back from the reviewers and the AE.

Best Regards,

Authors

---

### Decision · Action_Editor_srH3 · 2024-04-28

**Recommendation:** Accept with minor revision

**Comment:**

This submission investigates the relationship between neural collapse of pretrained networks and accuracy when those networks are transferred using a linear classifier or through fine-tuning. The submission shows that greater collapse as measured on downstream tasks is associated with greater transfer accuracy in a variety of settings. It proposes a “Skip Connection Layer Fine-tuning” strategy that fine-tunes only a single intermediate layer (the layer that exhibits the greatest collapse) with a skip connection to the output. Finally, it shows a result where neural collapse on the source task be positively or negatively correlated with transfer accuracy. Specifically, stronger augmentation leads to less collapsed representations that perform better on downstream tasks, but stronger adversarial training leads to less collapsed representations that perform worse on downstream tasks.

Reviewer D8Kd raised a concern that measuring NC on a downstream task is similar enough to classifier training that a correlation between the two is not particularly surprising. I generally agree—the performance of a classifier depends not only upon linear separability but also upon the difficulty of finding a good decision boundary from a finite dataset, which is in turn related to the amount of neural collapse. Reviewer aafv raised concerns regarding the thoroughness of the experimental setup. The revised paper includes some experiments in additional settings (e.g. neural collapse for CLIP models), and, although more thorough experimentation is possible, I doubt that it would radically change the findings. All reviewers raised issues regarding the clarity of parts of the paper, which the authors addressed in their revision.

I believe the revised submission satisfies TMLR’s acceptance criteria, and thus I recommend acceptance. Although not all results in the paper are surprising, the claims made are well-supported and the findings are likely to be of interest to some in TMLR’s audience.

For the final version of the paper, I request the following changes:

1. Reviewer D8Kd suggested revising the organization of the appendix to make it less scattered and moving some of these results into the main paper. The authors promised to do so, but didn’t want to make these changes before the final version since they would alter figure numbers. Please make sure to make the requested revisions.
2. There are some in-text citations that should be parenthetical citations (likely `\cite`s that should be `\citep`s) in the last bullet point of the introduction, the caption of Figure 2, and potentially other places.
3. Please double-check that ResNet34 and ViT-B are not switched between panels (a) and (b) of Figure 3.
4. Please clarify whether NC1 in Section 5 is computed on the training or test set.
5. There is at least one typo (“the NC1 of is too large”) on p. 12. Please do a final read-through to catch as many typos as you can.

**Audience:**

Yes.

**Claims And Evidence:**

Yes.